# REVIEWERTOO: SHOULD AI JOIN THE PROGRAM COMMITTEE? A LOOK AT THE FUTURE OF PEER REVIEW

## ABSTRACT

Peer review is the cornerstone of scientific publishing, yet it suffers from inconsistencies, reviewer subjectivity, and scalability challenges. We introduce **ReviewerToo**, a modular framework for studying and deploying AI-assisted peer review to complement human judgment with systematic and consistent assessments. ReviewerToo supports systematic experiments with specialized reviewer personas and structured evaluation criteria, and can be partially or fully integrated into real conference workflows. We validate ReviewerToo on a carefully curated dataset of 1,963 paper submissions from ICLR 2025, where our experiments with the `gpt-oss-120b` model achieves 81.8% accuracy for the task of categorizing a paper as accept/reject compared to 83.9% for the average human reviewer. Additionally, ReviewerToo-generated reviews are rated as higher quality than the human average by an LLM judge, though still trailing the strongest expert contributions. Our analysis highlights domains where AI reviewers excel (e.g., fact-checking, literature coverage) and where they struggle (e.g., assessing methodological novelty and theoretical contributions), underscoring the continued need for human expertise. Based on these findings, we propose guidelines for integrating AI into peer-review pipelines, showing how AI can enhance consistency, coverage, and fairness while leaving complex evaluative judgments to domain experts. Our work provides a foundation for systematic, hybrid peer-review systems that scale with the growth of scientific publishing.

## 1 INTRODUCTION

Major machine learning conferences such as ICLR and AAAI now receive (tens of) thousands of submissions every year, creating enormous pressure on the peer-review process. To cope with this scale, several venues begin experimenting with large language models (LLMs) as review assistants.[1] These early deployments demonstrate both promise (Liu & Shah, 2023; Petrescu & Krishen, 2022; Checco et al., 2021) and risk (Liang et al., 2024b; Latona et al., 2024): LLMs can generate consistent and scalable reviews, but they also produce superficial or misleading assessments that may erode confidence in the process.[2] Despite their visibility, such deployments remain one-off interventions constrained by conference timelines and are difficult to study in a reproducible manner.

A central challenge is that most reported outcomes of AI-assisted peer review remain anecdotal (even if large-scale), offering little scientific basis for best practices. Without systematic and reproducible evaluations, the community cannot determine where AI helps, where it harms, or how it might be responsibly integrated into review pipelines. Progress requires platforms that support controlled, transparent, and repeatable experiments—much like benchmarks have done for other areas of machine learning.

---

[1] E.g. AAAI 2026 (https://aaai.org/conference/aaai/aaai-26/instructions-for-aaai-26-reviewers/)

[2] https://www.nature.com/articles/d41586-025-00894-7

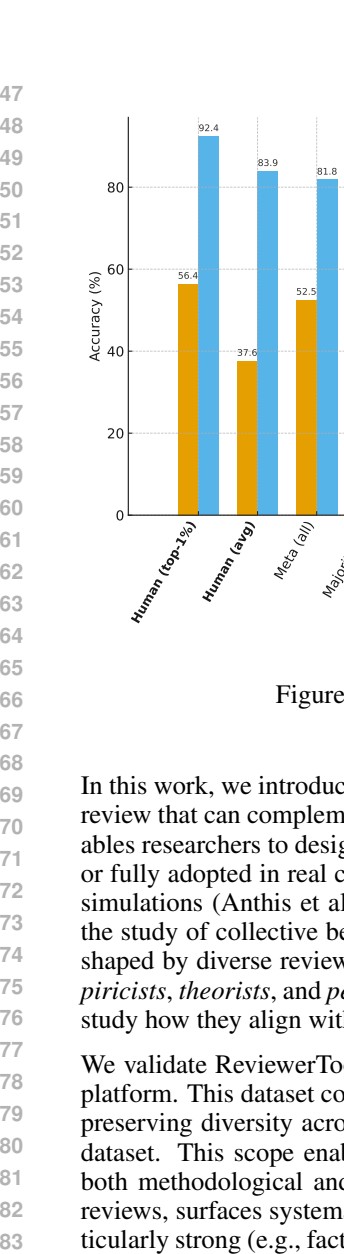

Figure 1: Performance of Different Reviewers on the ICLR-2k dataset.

In this work, we introduce **ReviewerToo**, a modular framework for studying and deploying AI-assisted peer review that can complement human judgment with systematic and consistent assessments. ReviewerToo enables researchers to design, test, and compare AI reviewers under standardized conditions, and it is partially or fully adopted in real conference workflows. We take inspiration from recent work on LLM-based social simulations (Anthis et al., 2025), which propose using language models as proxies for human subjects in the study of collective behavior. In this spirit, ReviewerToo treats peer review as a socio-technical process shaped by diverse reviewer roles, biases, and interactions. By instantiating reviewer personas–such as *empiricists*, *theorists*, and *pedagogical* reviewers–we use LLMs to simulate distinct reviewing philosophies and study how they align with human decisions.

We validate ReviewerToo on a curated dataset of ICLR 2025 submissions obtained from the OpenReview platform. This dataset consists of 1,963 papers sampled to balance acceptance and rejection decisions while preserving diversity across score ranges and decision categories. We refer to this dataset as the **ICLR-2k** dataset. This scope enables controlled yet realistic evaluation of AI-assisted reviewing at scale, yielding both methodological and empirical insights. Our analysis shows that ReviewerToo produces reasonable reviews, surfaces systematic biases across personas, and highlights dimensions where AI reviewers are particularly strong (e.g., fact-checking, literature coverage) or weak (e.g., assessing methodological novelty and theoretical contributions). These findings provide an evidence-based perspective on the opportunities and limitations of AI in peer review, moving beyond anecdote toward systematic study. In sum, this paper makes three contributions:

1. We conceptualize peer review as a socio-technical process and propose **ReviewerToo**, a modular framework for evaluating AI-assisted reviewing under controlled and transparent conditions.

2. We present a large-scale empirical study on the **ICLR-2k** dataset, analyzing the performance and biases of different reviewer personas and their alignment with meta-review outcomes.

3. We derive a set of guidelines for integrating AI into peer-review pipelines, informed by both quantitative performance metrics and qualitative analyses of reviewer behavior.

Together, these contributions provide a foundation for systematic and consistent integration of AI into the peer-review process.

## 2 BACKGROUND

**Challenges in Traditional Peer Review**  Peer review has long faced well-documented challenges, including reviewer fatigue, bias, and low inter-reviewer agreement (Cortes & Lawrence, 2021; Adam, 2025). Large-scale experiments at venues such as NeurIPS revealed that acceptance decisions can vary almost randomly (Cortes & Lawrence, 2021) and exhibit low inter-rater reliability. Combined with the rapid growth of submissions at top conferences (e.g., 11k+ and 25k+ at ICLR 2025 and NeurIPS 2025, respectively) and widespread reports of "reviewer fatigue," scalability has become a pressing concern (Adam, 2025).

**AI and LLMs as Peer-Review Assistants**  Recent advances in natural language processing (NLP) and large language models (LLMs) have spurred interest in using AI to assist peer review (Liang et al., 2024b; Tyser et al., 2024). Publishers and researchers have piloted systems for automated review generation, citation verification, fact-checking, and meta-review synthesis (Hossain et al., 2024). Surveys suggest that a substantial minority of reviewers are already using AI tools to speed up report writing, with some conferences estimating that 15–20% of reviews contain AI-assisted content (Latona et al., 2024; Naddaf, 2025). Empirical studies show mixed results: while LLM-generated reviews can be helpful according to authors, they also risk hallucinations and lack more in-depth judgment (Liang et al., 2023). Ongoing work thus emphasizes "AI-in-the-loop" designs, where models act as assistants for specific subtasks rather than as replacements for expert judgment (Idahl & Ahmadi, 2024; Liang et al., 2024a).

Despite this growing body of research, relatively little attention has been paid to *modeling reviewer diversity itself*. In practice, reviewers embody distinct philosophies—some emphasizing theoretical rigor, others empirical robustness, clarity of exposition, or long-term vision. Prior work on LLM-based social simulation shows that instantiating multiple role-specific agents can capture diverse perspectives in human decision processes (Sahakyan & AlShebli, 2025; Anthis et al., 2025). Inspired by this, we introduce REVIEWERTOO, a modular framework that explicitly models a plurality of reviewer personas. By simulating heterogeneous reviewer roles (e.g., "theorist," "empiricist," or "pedagogical"), our framework enables analysis not only of predictive accuracy against ground truth but also of the structure of inter-reviewer disagreement. This pluralistic design contributes both to practical peer-review augmentation and to the scientific understanding of reviewer dynamics. We include an extended literature review in Appendix B

## 3 SYSTEM OVERVIEW

REVIEWERTOO is a modular framework for studying and deploying AI-assisted peer review. It proceeds in a structured sequence: ingestion of the submitted manuscript, construction of a targeted literature review, generation of reviews by a diverse panel of reviewer agents, drafting of a consolidated rebuttal by an author agent, and finally a metareview that integrates the full record. The full workflow is shown in Figure 2.

We adopt a single-turn interaction protocol, in which each agent contributes once per stage (with the option for reviewers to issue one short post-rebuttal response). This choice reflects the conventions of many academic conferences, where reviewers typically provide a single review, authors submit one rebuttal, and only limited clarifications follow. While multi-turn deliberation could in principle be supported, our design prioritizes realism, and tractability, as LLMs have been shown to lose context in long, multi-turn discussions (Laban et al., 2025). We now discuss the different agents in our framework.

**The Literature Review Agent.**  For literature review, we use LitLLM (Agarwal et al., 2025), a retrieval-and-summarization agent proposed for automated literature review. Given a manuscript, LitLLM generates search queries and submits them to Semantic Scholar. Retrieved papers are ranked using a debate-based

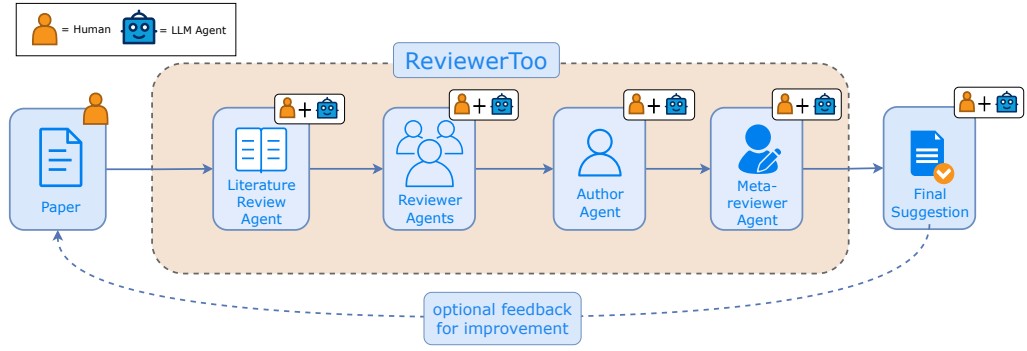

Figure 2: **The ReviewerToo Framework.** A paper passes through literature, reviewer, author, and meta-reviewer agents. The module design allows both humans and LLMs to participate at each stage, with optional feedback loops for iterative improvement.

method introduced in the original work, after which the top-$k$ candidates are selected. The agent summarizes these papers into a concise literature review that grounds subsequent reviewer, author, and metareviewer agents.

**Reviewer Agents.** Each reviewer agent receives the manuscript (converted to Markdown), an optional literature summary, and prompts encoding a specific reviewing persona or evaluation criteria. Reviewers generate structured assessments along axes commonly used in conference rubrics: a paper summary, explicit strengths and weaknesses, novelty, soundness, experimental validity, results/discussion quality, organization/presentation, and impact. For each dimension, reviewers must ground their judgments in either (i) explicit spans of the manuscript, or (ii) retrieved evidence from the literature summary. Additionally, the reviewer agent is also grounded in the official ICLR reviewer guidelines.[3] If no grounding can be located, the agent is rerun with stricter retrieval until a verifiable justification is produced. At the end of their report, reviewers provide a categorical recommendation from {Accept (Oral), Accept (Spotlight), Accept (Poster), Reject, Desk Reject}.

To surface complementary strengths and disagreements, we instantiate a diverse panel of personas. For brevity, we only mention a subset here, and we refer the reader to Table 5 for a more details:

- **Stance-based personas:** *critical* (reject-biased), *permissive* (accept-biased), and *default* (neutral).
- **Epistemic personas:** e.g., *theorist* (formal emphasis), *empiricist* (experimental rigor), *pedagogical* (clarity and exposition), and *pragmatist* (practical impact).
- **Stylized personas:** caricatured reviewer archetypes such as *visionary* (long-term potential), *probabilistic* (uncertainty reasoning), and *impact-driven* (field-level relevance).

**Author Agent.** The author agent takes the manuscript, the full set of reviewer reports, and the literature summary as input. It generates a consolidated rebuttal that addresses the most severe criticisms, clarifies potential misunderstandings, and, when appropriate, proposes concrete revisions such as releasing code or adding ablation studies. The rebuttal must explicitly cite either reviewer claims or relevant literature, ensuring that clarifications are verifiable rather than speculative. Rebuttals are stored per review configuration to facilitate analysis.

**Metareviewer Agent.** The metareviewer integrates all reviewer reports, the author rebuttal, and any optional post-rebuttal reviewer responses. Its role is to synthesize consensus while controlling for reviewer disagreement and bias. Concretely, it: (1) summarizes reviewer stances and scores pre-rebuttal, (2) identifies

---

[3]https://iclr.cc/Conferences/2025/ReviewerGuide

Table 1: Main Results on ICLR-2k Dataset. Best results (per block, per column) are in bold.

| Agent | 5-way | | | | 2-way | | | | ELO$^{\uparrow}$ |
|---|---|---|---|---|---|---|---|---|---|
| | P$^{\uparrow}$ | R$^{\uparrow}$ | F$^{\uparrow}$ | A$^{\uparrow}$ | P$^{\uparrow}$ | R$^{\uparrow}$ | F$^{\uparrow}$ | A$^{\uparrow}$ | |
| **Single ReviewerToo Agents** | | | | | | | | | |
| Theorist | 31.0 | 24.0 | 22.6 | 45.9 | 72.1 | **72.1** | **71.9** | **71.9** | 1463 |
| Pedagogical | 27.1 | 23.0 | 21.0 | **51.7** | 72.9 | 68.9 | 68.3 | 70.3 | 1256 |
| Empiricist | 32.5 | 22.5 | 20.6 | 50.7 | 69.7 | 66.1 | 65.3 | 67.6 | **1558** |
| Critical | 12.5 | 17.0 | 11.9 | 49.6 | **76.8** | 50.1 | 35.0 | 53.6 | 423 |
| Permissive | 10.5 | 16.8 | 7.5 | 19.1 | 73.3 | 50.3 | 32.4 | 46.8 | 880 |
| Default | 26.7 | 24.5 | 21.8 | 43.3 | 72.4 | 71.5 | 70.5 | 70.6 | 1136 |
| DeepReview-14B | 23.4 | 21.9 | 20.6 | 37.3 | 70.4 | 63.7 | 62.4 | 62.5 | 1117 |
| Liang et al. (2024b) | **37.6** | **28.3** | **23.5** | 42.5 | 72.9 | 71.9 | 70.9 | 71.0 | 1202 |
| **Reviewer Ensembles** | | | | | | | | | |
| Majority (Top-3) | 30.5 | 28.5 | 25.9 | 52.0 | 73.1 | 70.0 | 69.8 | 71.2 | – |
| Majority (all) | 30.7 | 30.0 | 27.9 | 49.2 | 75.1 | 75.2 | 75.1 | 75.1 | – |
| Average (all) | 32.5 | 26.4 | 22.7 | 42.2 | 68.6 | 65.0 | 60.3 | 64.8 | – |
| Meta w/ Top-3 | 28.6 | 32.1 | 27.1 | 49.5 | 74.2 | 76.3 | 73.4 | 74.8 | 1329 |
| Meta w/ Top-3+Base | 26.7 | 26.1 | 19.5 | 30.4 | 74.7 | 63.6 | 57.3 | 61.2 | 1154 |
| Meta (all) | **32.1** | **32.4** | **28.1** | **52.5** | **79.3** | **80.1** | **79.3** | **81.8** | **1657** |
| **Supervised Baselines** | | | | | | | | | |
| XGBoost (Bert) | 12.9 | 20.0 | 14.4 | 20.0 | 59.0 | 55.8 | 44.3 | 55.8 | – |
| XGBoost (tfidf) | 17.4 | 21.4 | 17.4 | 21.4 | 70.4 | 63.7 | 58.2 | 63.7 | – |
| Bert FT | 25.7 | 26.4 | 22.5 | 22.4 | 84.2 | 29.1 | 43.24 | 65.43 | – |
| **Human Baselines** | | | | | | | | | |
| Human (avg) | 15.2 | 12.4 | 13.7 | 37.6 | 85.2 | 84.1 | 83.8 | 83.9 | 540 |
| Human (top-1%) | 31.5 | 30.4 | 29.7 | 56.4 | 93.7 | 91.9 | 90.4 | 92.4 | 1316 |

common strengths and weaknesses, (3) evaluates rebuttal effectiveness, (4) tracks stance shifts post-rebuttal, and (5) highlights lingering concerns or unresolved disagreements.

To avoid being swayed by overly negative or idiosyncratic reviewers, the metareviewer includes a fact-checking module. This module verifies reviewer-stated claims against both the manuscript and the literature summary, discarding unsupported statements. Each fact is also assigned a significance score, indicating its weight in shaping the final decision. The final metareview thus reflects a combination of consensus synthesis, rebuttal analysis, and fact-weighted evidence assessment. Notably, the metareviewer agent is also grounded in the official Area Chair guidelines from the ICLR.[4] We include the implementation details of the system in Appendix E and include our prompts in the supplementary material.

---

[4]https://iclr.cc/Conferences/2025/ACGuide

## 4 EXPERIMENTAL SETUP

### 4.1 DATASETS

All experiments are conducted on a dataset derived from ICLR 2025 submissions to the OpenReview platform. From the full pool of 11,672 submissions, we curate a stratified subset of 1,963 papers, which we refer to as the **ICLR-2k** dataset. We focus on this subset for all reported results, as it enables balanced coverage of decision categories and controlled ablation studies. Each submission is annotated with the official conference decision, which serves as the ground truth for both five-way and binary evaluations. We consider five categories: *Accept (Oral)*, *Accept (Spotlight)*, *Accept (Poster)*, *Reject*, and *Desk Reject*. Withdrawn papers are merged into the *Reject* category, while *Desk Reject* is preserved separately to test the system's ability to detect incomplete or rule-violating submissions. To ensure representativeness, we first rank all ∼12k submissions (213 orals, 380 spotlights, 3115 posters, 7894 rejected, 70 desk rejected) by average reviewer score and then sample proportionally across the score distribution. Specifically: (i) For *Accepted (Poster)*, we select 300 of 3,115 submissions, sampling evenly from the top, middle, and bottom thirds of the ranked list; (ii) For *Reject*, we include 500 of 5,019 submissions using the same stratification, and add 500 randomly sampled withdrawn papers; (iii) For *Accepted (Oral)*, *Accepted (Spotlight)*, and *Desk Reject*, we include all available cases. This design balances acceptance and rejection while preserving diversity across decision types and score ranges.

### 4.2 BASELINES

We evaluate REVIEWERTOO on multiple baselines ranging from trivial heuristics to human-derived signals. Our baselines fall into four groups: **(1) Supervised Baselines.** We include three supervised baselines where we train an XGBoost classifier with TF-IDF features (**XGBoost (tfidf)**), and XGBoost classifier with frozen BERT embeddings as features (**XGBoost (bert)**), and BERT classifier finetuned on the dataset (**Bert FT**). **(2) Single-agent reviewers.** represents a reviewer agent conditioned on ICLR conference instructions and literature review. **(3) Reviewer ensembles.** We test whether diversity and aggregation improve fidelity. **(a) Majority vote:** across all reviewer personas. **(b) Extremal ensembles:** combining permissive and critical personas to probe systematic bias. **(c) Metareviewer aggregation:** synthesizing all reviews and rebuttals into a calibrated consensus. **(4) Human Reviewers.** We benchmark the performance of human reviewers. Specifically, we report: **(a) Human (avg):** performance of all the human reviewers. **(b) Human (top-1%)** We use the ELO scores to select the top-1% human reviewers and report their performance. Notably, since human reviewers only provide scores to a paper, we perform threshold-based conversion to obtain their predictions. For 2-way (binary) case, we treat scores $> 5$ as accept and scores $< 5$ as rejects. For the 5-way case, we use the following thresholds: {Accept (Oral): 7.8, Accept (Spotlight): 7.4, Accept (Poster): 6.05, Reject: 4.11, Desk Reject: 0.54} [5].

To isolate the contribution of structured protocols, we ablate on the different conditioning variables: **(a)** $\phi$**:** represents a reviewer agent without any conditioning on conference instructions, or literature review, or rebuttal. It only takes as input the manuscript and responds according to its base personality imbued in the system prompt. **(b) CI:** adds ICLR reviewers guidelines for the reviewer agents and area chair guidelines for metareviewer agent in addition to the persona-specific instructions. **(c) RB:** extends conference conditioning with an author rebuttal and one round of reviewer response. **(d) LitLLM:** further incorporates external retrieval and summarization (LitLLM). This sequence reflects a controlled ablation from bare-bones to fully contextualized reviewing.

Together, these baselines span uninformed heuristics, isolated reviewer agents, structured multi-agent protocols, ensembles, and human artifacts. This progression allows us to evaluate two complementary questions:

---

[5]thresholds for each class is determined based on the average score of papers in that class in the ICLR-2k dataset

**(1)** how effective LLMs are as reviewers in absolute terms, and **(2)** which design choices most narrow the gap to human decision-making.

### 4.3 EVALUATION METRICS

We assess REVIEWERTOO along multiple axes that capture predictive accuracy, reviewer agreement, review quality, and rebuttal helpfulness. We evaluate alignment with real conference decisions by measuring both the 5-way classification performance (Oral, Spotlight, Poster, Reject, Desk Reject) and the binary Accept/Reject task; we report macro-averaged **Precision**, **Recall**, and **F1**, with macro averaging across classes $c$. We also report overall **Accuracy**, and **False Positive Rate** (for binary task). We quantify consistency among reviewers and with the metareviewer. For two reviewers $i, j$, we compute **Cohen's** $\kappa$

**Review quality.** We assess the quality of review text through LLM-based judgments. We conduct large-scale pairwise comparisons where an LLM acts as the judge. For each paper, two reviews are shown side by side and evaluated along five axes: (1) *Depth* of engagement with the paper's methodology and arguments; (2) *Actionability*, i.e., whether weaknesses are paired with concrete suggestions and is the feedback constructive; (3) *Summary*, i.e. whether the agent identified strengths and weakness of the paper in a balanced manner; (4) *Clarity*, reflecting readability, structure and professionalism; and (5) *Helpfulness* of the review to the author. The judge assigns a win, loss, or draw outcome to each review. From the full set of pairwise outcomes we compute an **ELO rating** per system, which is a method for calculating the relative skill levels of players in zero-sum games such as chess or esports. In our case, different reviewer agents are "players". We include the complete protocol in Appendix C.

## 5 RESULTS

**Reviewer Performance.** Table 1 reports the performance of REVIEWERTOO agents, supervised baselines, and human references. Figure 1 visualizes the F1 Scores from that table. Among single-agent reviewers, the EMPIRICIST, PEDAGOGICAL, and THEORIST personas achieve the strongest overall performance on the 5-way classification task, with the EMPIRICIST attaining the highest precision (32.5) while THEORIST secures the best F1 score (22.6). In terms of binary accept/reject accuracy, these reviewers approach 70% accuracy, narrowing the gap to human baselines. Ensembling further boosts performance: majority voting improves stability, while the metareviewer aggregation ("Meta

Table 2: Ablation Results for conference instructions (CI), LitLLM, and rebuttal (RB).

| Agent (Configuration) | F1$^\uparrow$ | ELO$^\uparrow$ | FPR$^\downarrow$ | FNR$^\downarrow$ |
|---|---|---|---|---|
| Theorist ($\phi$) | 67.4 | 1371 | 96.1 | 42.0 |
| +CI | 69.9 | 1422 | 73.6 | 71.3 |
| +CI+LitLLM | 71.9 | 1463 | 76.0 | 68.4 |
| +CI+RB | 63.8 | 1299 | 85.6 | 46.3 |
| +CI+LitLLM+RB | 63.6 | 1195 | 88.0 | 44.5 |
| Pedagogical ($\phi$) | 75.5 | 1345 | 70.3 | 80.5 |
| +CI | 70.5 | 1256 | 45.4 | 90.9 |
| +CI+LitLLM | 68.2 | 1216 | 48.7 | 88.8 |
| +CI+RB | 61.9 | 1103 | 78.9 | 49.9 |
| +CI+LitLLM+RB | 63.0 | 1122 | 76.3 | 49.8 |
| Empiricist ($\phi$) | 69.1 | 1502 | 84.0 | 55.9 |
| +CI | 64.8 | 1427 | 43.8 | 86.3 |
| +CI+LitLLM | 70.7 | 1558 | 45.4 | 87.1 |
| +CI+RB | 59.7 | 1316 | 75.7 | 47.8 |
| +CI+LitLLM+RB | 60.4 | 1332 | 73.5 | 48.5 |

(all)") outperforms both single-agent and majority ensembles across all metrics, reaching 32.1 precision, 32.4 recall, and 28.1 F1 on the 5-way task, and 81.8% accuracy on the binary task. This model also achieves the strongest ELO score of 1657, surpassing all other agents and aligning closely with the top-1% human baseline.

**Error Analysis via Confusion Matrices.** Figures 5–6 present normalized confusion matrices for each agent. We observe consistent difficulty in distinguishing between "oral" and "spotlight" accept decisions across nearly all personas, indicating sensitivity to fine-grained acceptance tiers. Expectedly, the PERMISSIVE persona over-predicts acceptance decisions, while the CRITICAL persona strongly favors rejection. By contrast, the EMPIRICIST and PEDAGOGICAL show more balanced error profiles, though they still over-predict rejections relative to ground truth. These error modes highlight both biases induced by personas and systematic challenges in conference calibration.

**Reviewer Agreement.**    We quantify inter-reviewer consistency using Cohen's $\kappa$ (Figure 3). Agreement levels vary substantially across personas: MAJORITY and DEFAULT show moderate alignment ($\kappa \approx 0.5$), while PERMISSIVE and CRITICAL show near-zero or even negative agreement with other reviewers, underscoring their extremal tendencies. Human reviewers exhibit low to moderate agreement with LLM reviewers ($\kappa \approx 0.1$–$0.2$), consistent with known levels of disagreement in real peer review. Ensembles such as MAJORITY and META yield higher agreement with ground truth, validating aggregation as a stabilizing mechanism.

**Review Quality and ELO.**    Beyond predictive accuracy, we assess review quality through LLM-based pairwise judgments, aggregated with ELO ratings. The META (ALL) agent again dominates, achieving the highest ELO of 1657. Among single-agent reviewers, the EMPIRICIST leads with 1558, while the PEDA-GOGICAL and THEORIST trail but still outperform most supervised baselines. Interestingly, human reviewers exhibit a striking disparity: the average human ELO is very low (540), yet the top 1% of human reviewers achieve an ELO of 1316, comparable to the best single-agent reviewers. At the same time, both average and top-1% humans maintain strong binary F1 performance (83.8 and 90.4, respectively). The textual reviews they produce are often less helpful by the criteria used in our LLM-as-judge framework–particularly with respect to actionability and usefulness to authors. These findings reinforce the potential of structured protocols, diversity, and meta-reviewing to not only improve decision alignment but also to generate more constructive review text.

**Comparison with Supervised Baselines.**    Supervised baselines such as XGBoost and BERT fine-tuning achieve modest predictive performance, with binary F1 scores ranging from 43.2 to 65.3. In contrast, RE-VIEWERTOO agents not only match or exceed these baselines in decision accuracy but also generate substantive reviews that achieve competitive or superior ELO ratings. Unlike humans, who remain strong on both axes–achieving high binary F1 performance while also producing text that can be judged for quality–supervised models cannot bridge the gap between decision fidelity and helpful feedback. This underscores the unique advantage of structured reviewer agents in combining predictive alignment with author-facing utility.

**Ablation Studies.**    Table 2 examines the impact of conditioning variables. Removing conference instructions systematically reduces both F1 and ELO, indicating their critical role in reviewer fidelity. For example, the EMPIRICIST with full conditioning (+CI+LitLLM) achieves the highest ELO (1558), whereas ablations removing literature grounding drop performance sharply (e.g., ELO $\leq 1332$). Interestingly, PEDAGOGICAL shows the highest raw F1 score (75.5) in its base persona, though its ELO is lower, suggesting less consistent quality under comparative evaluation. Overall, ablations confirm the complementary value of structured conference context, literature retrieval. We also see that the F1 score for all the reviewer agents drops post rebuttal. Upon closer inspection, we note that this is due to an increase in false positives post discussion while both false negatives and true negatives decrease (compare CI+LitLLM v/s CI+LiLLM+RB rows in Table 2). This hints towards sycophantic tendencies of LLMs to accept papers after reading the rebuttals, as from their point-of-view, they come from real humans (Kim & Khashabi, 2025; Sun & Wang, 2025; Cheng et al., 2025). The increase in false positives can also be visually seen confusion matrices for different reviewer agents before and after discussion (see Figure 5 v/s Figure 4 & Figure 6 v/s Figure 7). We have also attached

**Qualitative Examples.**    We show the reviews generated for two papers, one from the accept category (PaperID: 6Mxhg9PtDE) and one from the reject category (PaperID: j7b4mm7Ec9). For each paper, we compare the reviews written by the top-3 reviewer agents (pedagogical, empiricist, and theorist) with a human review (we only include one human review but the links to the full threads are provided). Full examples are provided in Appendix D. **For the accepted paper**, we observe that the reviewer agents not only reach the same accept decision as the human reviewers but also produce comprehensive and well-structured feedback, covering key aspects such as novelty, clarity, and experimental rigor. This consistency suggests that, for strong submissions, the agents are capable of recognizing high-quality research and articulating reasoned justifications similar to human reviewers. **The rejected paper** presents a more nuanced case.

Interestingly, while all human reviewers gave positive (accept) scores, the meta-reviewer's final decision was to reject the paper. Notably, some of our reviewer agents also assigned reject decisions for similar reasons as the metareviewer (revolving around the true robustness of the proposed approach). This alignment with the final decision, despite divergence from the human reviewers' initial scores, indicates that the agents can independently identify underlying weaknesses in a paper that might be overlooked in human assessments. Such behavior demonstrates the potential of AI reviewer agents to provide balanced, critical evaluations and to meaningfully contribute to peer review deliberations.

**Summary.** Taken together, these results demonstrate that REVIEWERTOO can reasonably approximate human-level decision making, especially when aggregating diverse reviewers through metareview protocols. Single-agent personas exhibit distinctive biases, but structured ensembles yield both higher predictive accuracy and higher judged review quality. Agreement analysis highlights persistent reviewer variance, mirroring human peer review. Finally, ablation studies confirm that conference conditioning, rebuttals, and literature access are each essential to closing the gap with human reviewers.

## 6 DISCUSSION

Our experiments on the ICLR-2k dataset provide a first large-scale analysis of how LLM-based reviewer agents perform relative to humans, supervised classifiers, and ensemble protocols. The results reveal both opportunities and limitations of using AI in peer review. Here, we synthesize these findings into broader lessons and propose practical guidelines for integrating AI into peer-review pipelines.

**AI reviewers approximate but do not replace humans.** The results show that single-agent reviewer personas achieve accuracy close to 70% on the binary accept/reject task, narrowing the gap with human baselines. However, their five-way performance remains substantially lower, and confusion matrices highlight consistent difficulty in distinguishing fine-grained acceptance tiers (e.g., oral vs. spotlight). This suggests that LLM reviewers can approximate coarse-grained decision making, but conference-level calibration still requires human expertise. Importantly, human reviewers maintain higher binary F1 scores, underscoring their ability to holistically evaluate paper quality.

**Ensembles and metareviewing stabilize and improve fidelity.** Our ensemble protocols consistently outperform single-agent reviewers, with the META (ALL) agent achieving the strongest results across accuracy, F1, and ELO. Aggregating multiple perspectives reduces individual biases (e.g., permissive vs. critical personas) and yields more reliable decision-making. This mirrors existing human peer review, where program committees rely on multiple reviews and meta-review synthesis to mitigate individual noise. Our findings indicate that metareviewing is a crucial design principle for AI-assisted peer review.

**Quality of review text remains a challenge.** ELO ratings highlight that while reviewer agents can generate more constructive feedback than supervised baselines, the quality of their review text is not always aligned with human expectations. Average human reviews perform poorly under ELO, suggesting that even human-authored text often fails on criteria such as actionability and helpfulness to authors. At the same time, the top 1% of human reviewers achieve high ELO, showing that exemplars exist. These results caution that AI reviews should be seen as complements–providing structured, constructive feedback–rather than replacements for nuanced human judgment.

**Rebuttals introduce sycophancy risks.** Ablation studies reveal that performance systematically drops after rebuttal rounds due to sycophantic tendencies of LLMs: they may defer excessively to rebuttals without maintaining independent judgment. This highlights a need for careful design of how LLM reviewers handle author feedback. Safeguards, such as explicit calibration instructions or adversarial prompting, may be required to prevent performance degradation in rebuttal phases.

**Reviewer agreement mirrors human inconsistency.** Pairwise Cohen's $\kappa$ shows that LLM reviewers vary substantially in their agreement, with some personas (e.g., permissive, critical) diverging strongly from

others. This echoes longstanding challenges in human peer review, where reviewer disagreement is common. Our findings suggest that AI reviewers will not eliminate variance in peer review but can be structured to reduce it through ensembles and consensus protocols.

### 6.1 GUIDELINES FOR INTEGRATING AI INTO PEER REVIEW

From these quantitative and qualitative findings, we propose a set of guidelines for integrating AI into peer-review pipelines:

1. **Use AI reviewers as complements, not replacements.** LLM reviewers can provide scalable, structured feedback and approximate decision accuracy, but final judgments should remain with humans, particularly for borderline and high-stakes decisions.

2. **Prioritize ensemble protocols.** Single-agent reviewers exhibit strong biases; aggregation through majority voting or metareviewing produces more reliable and fair outcomes. AI systems in peer review should default to ensemble-based designs.

3. **Incorporate structured conditioning.** Conference-specific guidelines, literature retrieval, and rebuttal phases each add value, but must be carefully balanced to avoid overfitting or sycophancy. Conditioning improves fidelity, but uncritical incorporation of rebuttals can degrade performance.

4. **Evaluate not just accuracy, but also review quality.** Our ELO analysis highlights that decision fidelity alone is insufficient; reviews must also be actionable and useful to authors. AI reviewers should be explicitly optimized for feedback quality as well as predictive accuracy.

5. **Human-AI collaboration as the design goal.** The stark gap between average and top-1% human reviewers suggests a role for AI in "raising the floor": providing consistent, constructive baseline reviews that can complement and support human judgment, rather than competing with it.

6. **Mitigate bias and disagreement through protocol.** Extremal personas can systematically over- or under-predict acceptance. Careful design of reviewer ensembles and meta-review synthesis is essential to reduce variance and ensure fairness in outcomes.

## 7 CONCLUSION

Peer review is central to scientific publishing but remains plagued by inconsistency, subjectivity, and scalability limits. We introduced REVIEWERTOO, a modular framework for AI-assisted peer review that leverages structured reviewer personas, ensemble protocols, and systematic evaluation. On the ICLR-2k dataset, LLM reviewers approached human-level decision accuracy—especially under metareviewing—and produced reviews often judged more constructive than the human average. Yet challenges such as fine-grained calibration, susceptibility to sycophancy during rebuttals, and variable persona agreement highlight the continued need for human expertise. From these results we propose guidelines for hybrid peer review: deploy AI reviewers as complements rather than replacements, prioritize ensembles and meta-review protocols, condition agents with structured context, and optimize for both review quality and decision fidelity. With such workflows, AI can enhance consistency, coverage, and fairness, while humans provide the nuanced judgments essential for advancing science.

## 8 ETHICS STATEMENT

This work involves the use of publicly available data from the OpenReview platform, which hosts peer review information for academic conferences. We strictly adhered to OpenReview's terms of use and community guidelines in collecting and analyzing this data. All data used were publicly accessible at the time

of collection and we did not pass any personally identifiable information to the LLM beyond what was intentionally made public by authors or reviewers.

The goal of this research is to improve the transparency, scalability, and fairness of the peer review process. Our experiments are designed to complement, not replace, human reviewers, with the intent of assisting editorial processes and studying the potential of AI tools in structured academic evaluation. We emphasize that ReviewerToo is meant for research and controlled integration scenarios, and not for unsupervised or fully automated decision-making in academic publishing.

Finally, the use of large language models in review generation and evaluation was conducted with attention to ethical implications, including the risks of bias propagation, overreliance on model outputs, and possible reinforcement of systemic inequities. We provide concrete guidelines and limitations in our discussion to promote responsible adoption of AI-assisted review systems.

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

## A  LLM Usage

We have used LLMs to improve the text. Specifically, we have use chatGPT to improve the language of some paragraphs and we have used LitLLM to retrieve relevant works.

## B  Extended Literature Review

The exponential growth of scientific submissions has placed an unprecedented strain on the peer review system, leading to reviewer fatigue, inconsistency, and scalability challenges. Recent advancements in Large Language Models (LLMs) offer a potential solution, yet their integration into the "Program Committee" remains a subject of intense debate. This review synthesizes the current landscape of AI in scholarly publishing, motivating the proposed *ReviewerToo* framework.

### B.1  Perceptions and the Role of AI in Peer Review

The academic community is currently divided on the integration of AI into the review process. While some view it as a necessary evolution to handle volume, others fear the loss of human nuance.

**Community Sentiment and Adoption**  Surveys and opinion pieces highlight a spectrum of attitudes. Some researchers argue that AI can "democratize" knowledge creation and streamline workflows (Sarker et al., 2024; Ravn Sørensen, 2024; Švab et al., 2023). However, significant skepticism remains regarding the "death of the human reviewer" and the potential for an "AI ouroboros" where AI reviews AI content (Mollaki, 2024; Poole & Todd-Diaz, 2025; Mamassian, 2025).

Empirical studies on researcher perceptions reveal a cautious willingness to adopt tools for drafting and checking, but hesitation regarding evaluative judgments (Giray, 2024; Ebadi et al., 2025; Banks, 2024; Ali & Shaban, 2025; Ng et al., 2025a;b). Several studies emphasize the need for human supervision to maintain integrity (Seghier, 2025; Crawford et al., 2024; Renata & Lee, 2025).

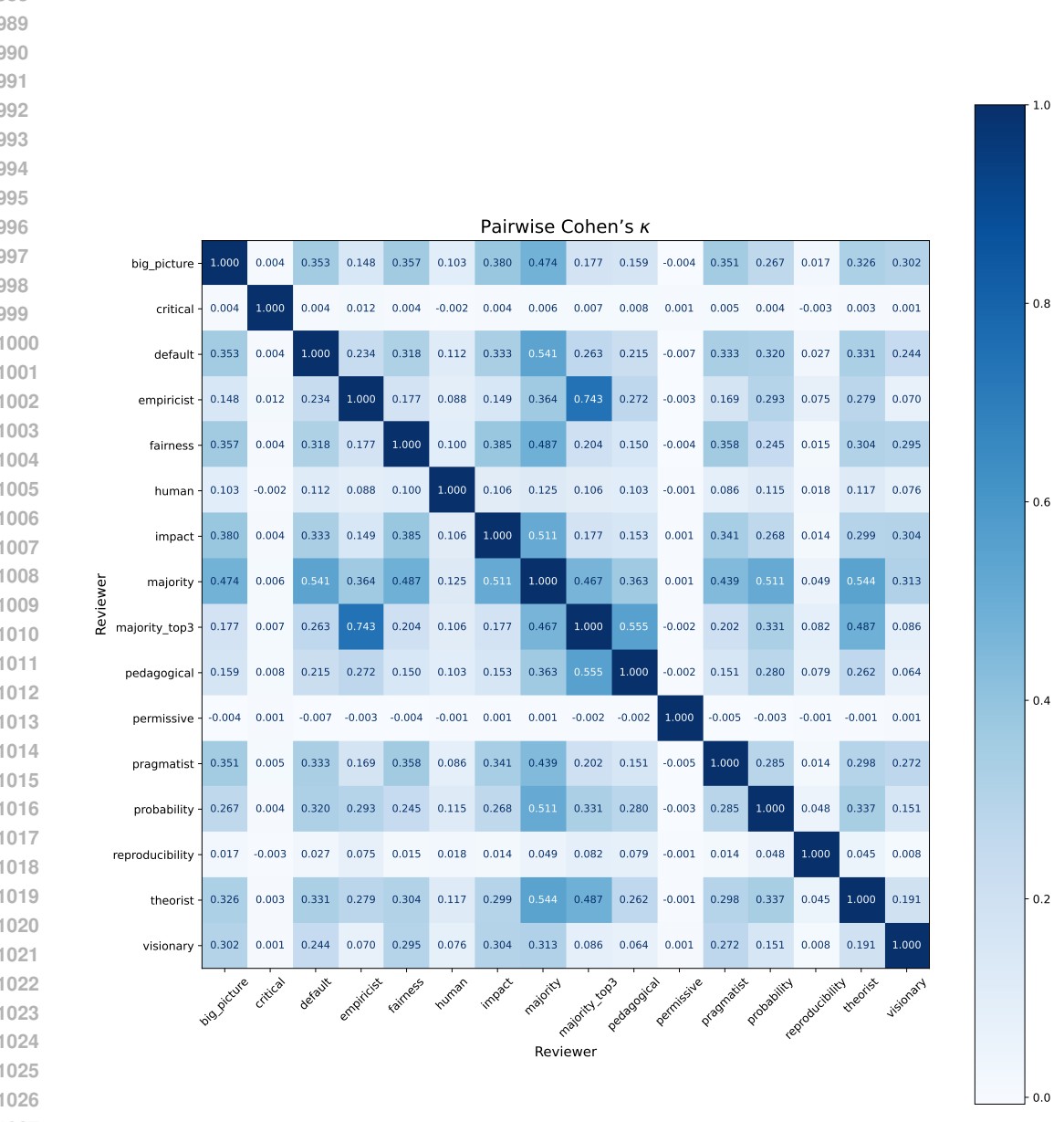

Figure 3: Pairwise Cohen's $\kappa$ for different types of reviewers

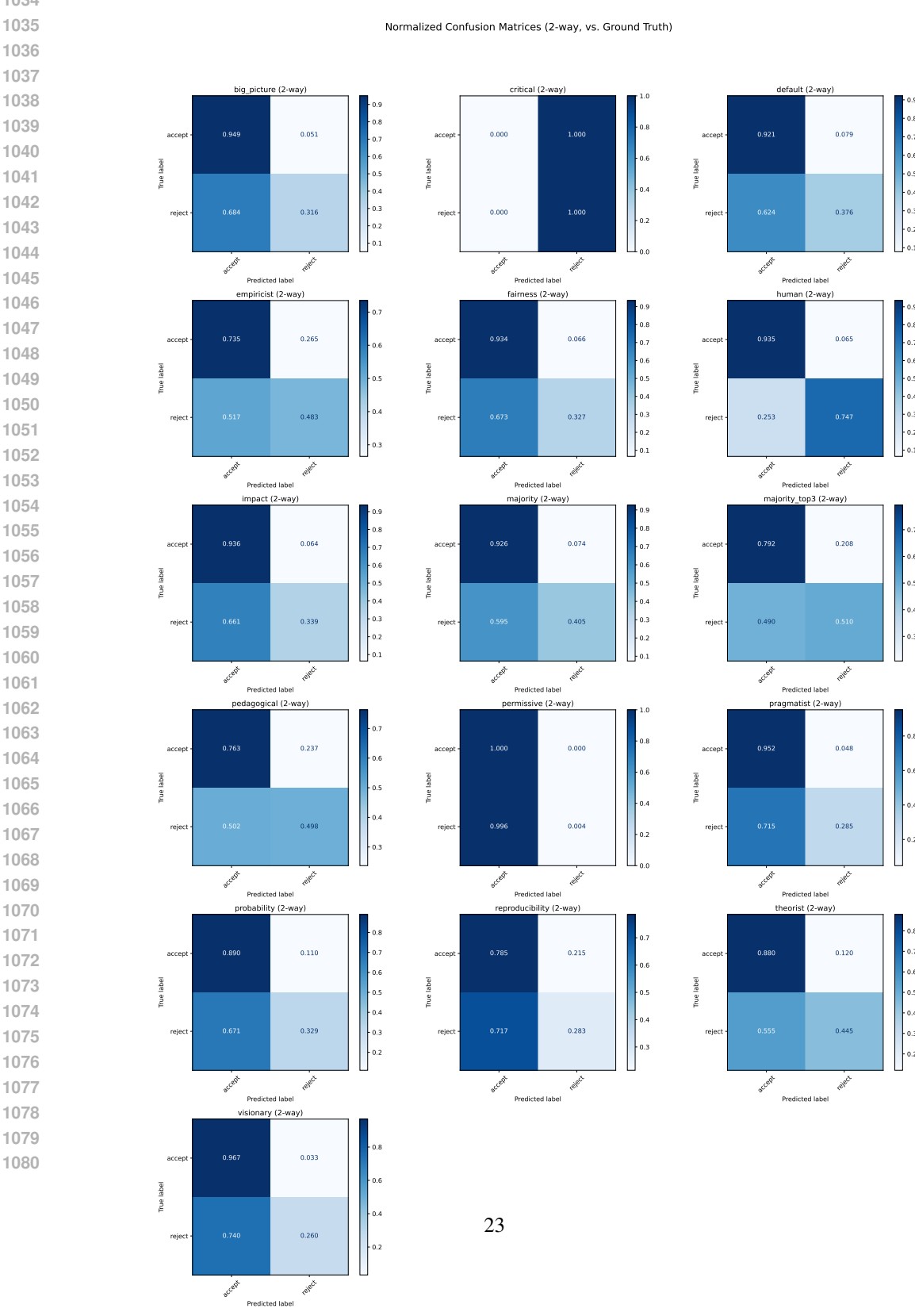

Figure 4: Confusion Matrices for binary Classification Task Post-Discussion

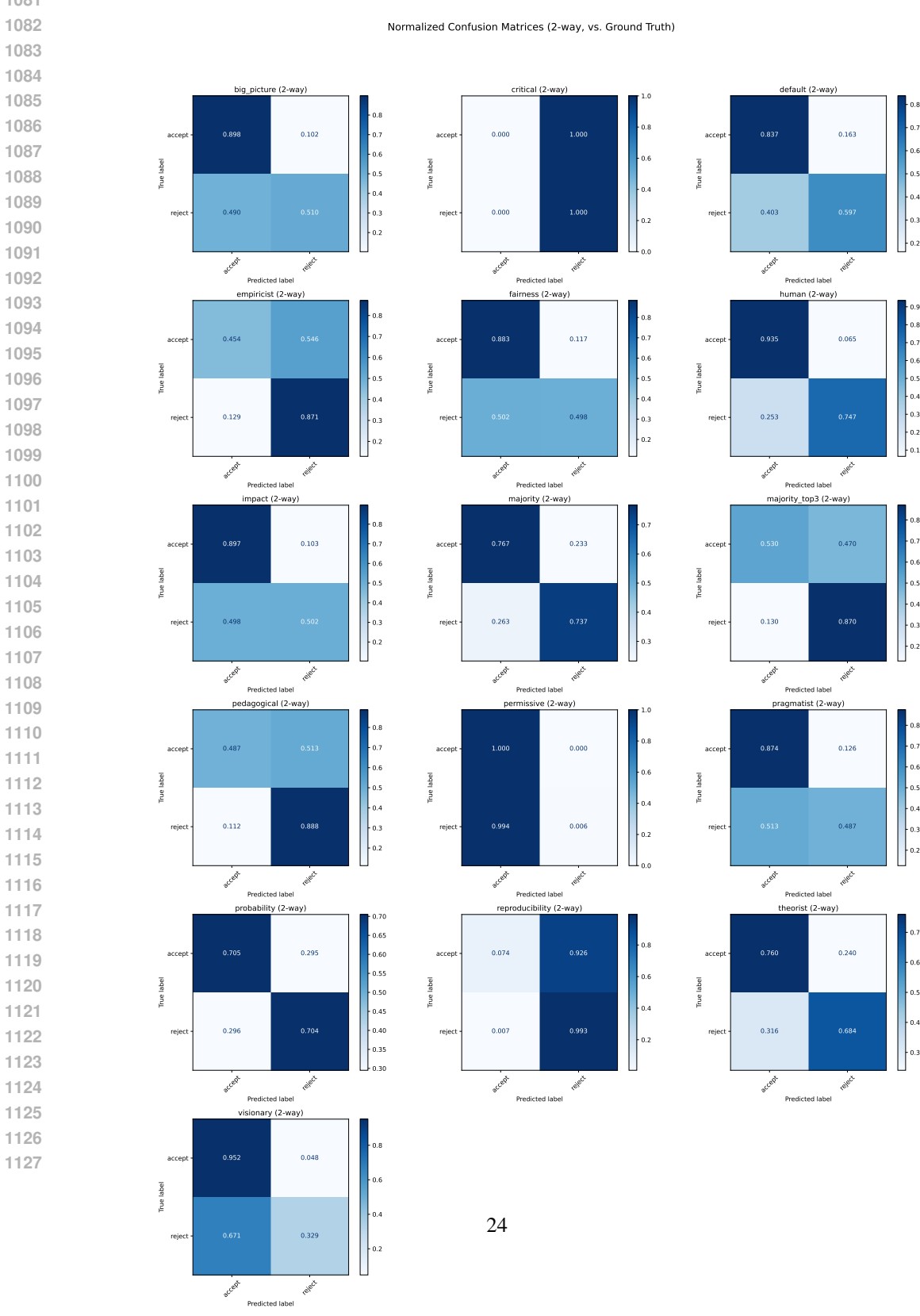

Figure 5: Confusion Matrices for binary Classification Task

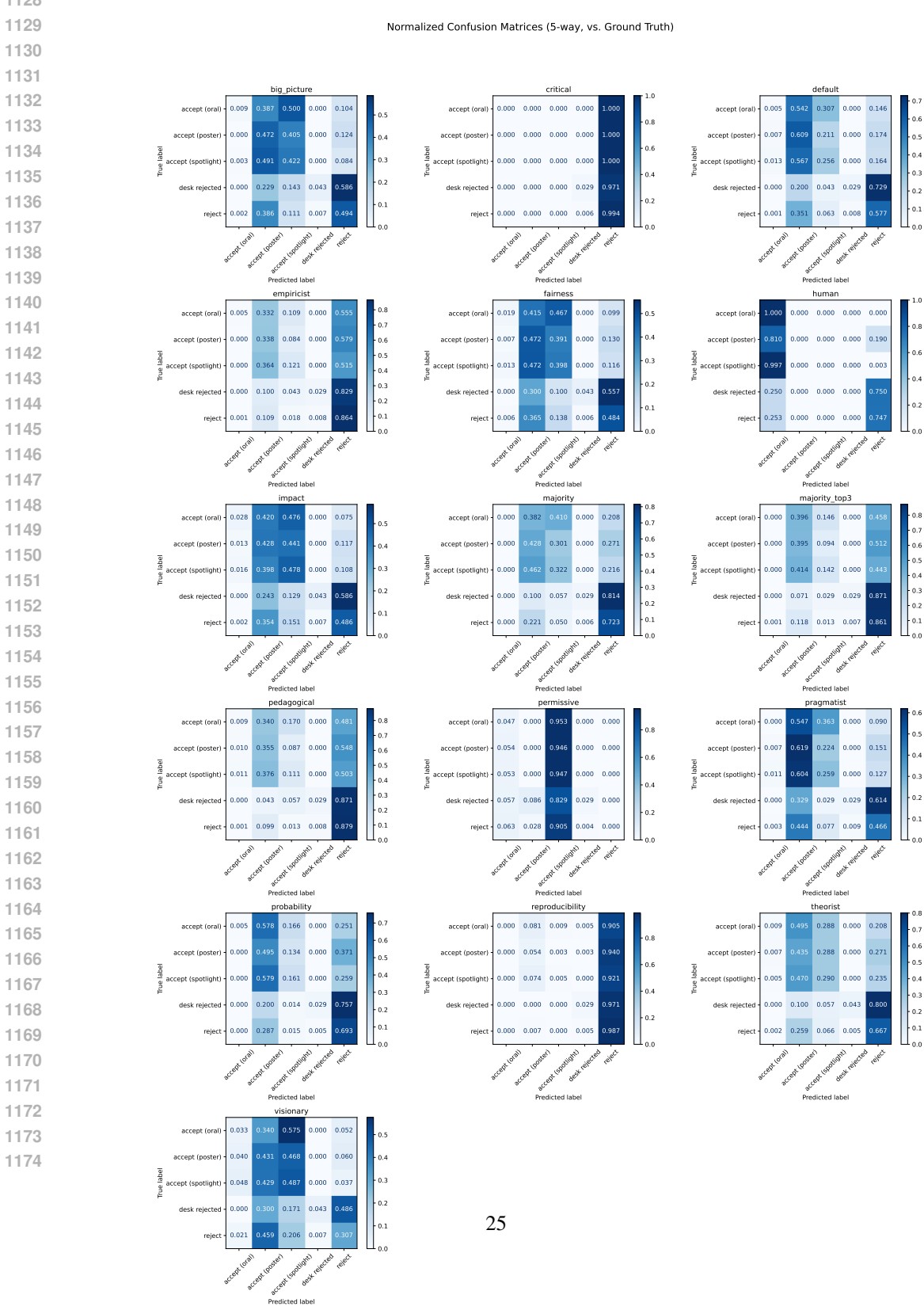

Figure 6: Confusion Matrices for 5-way Classification Task

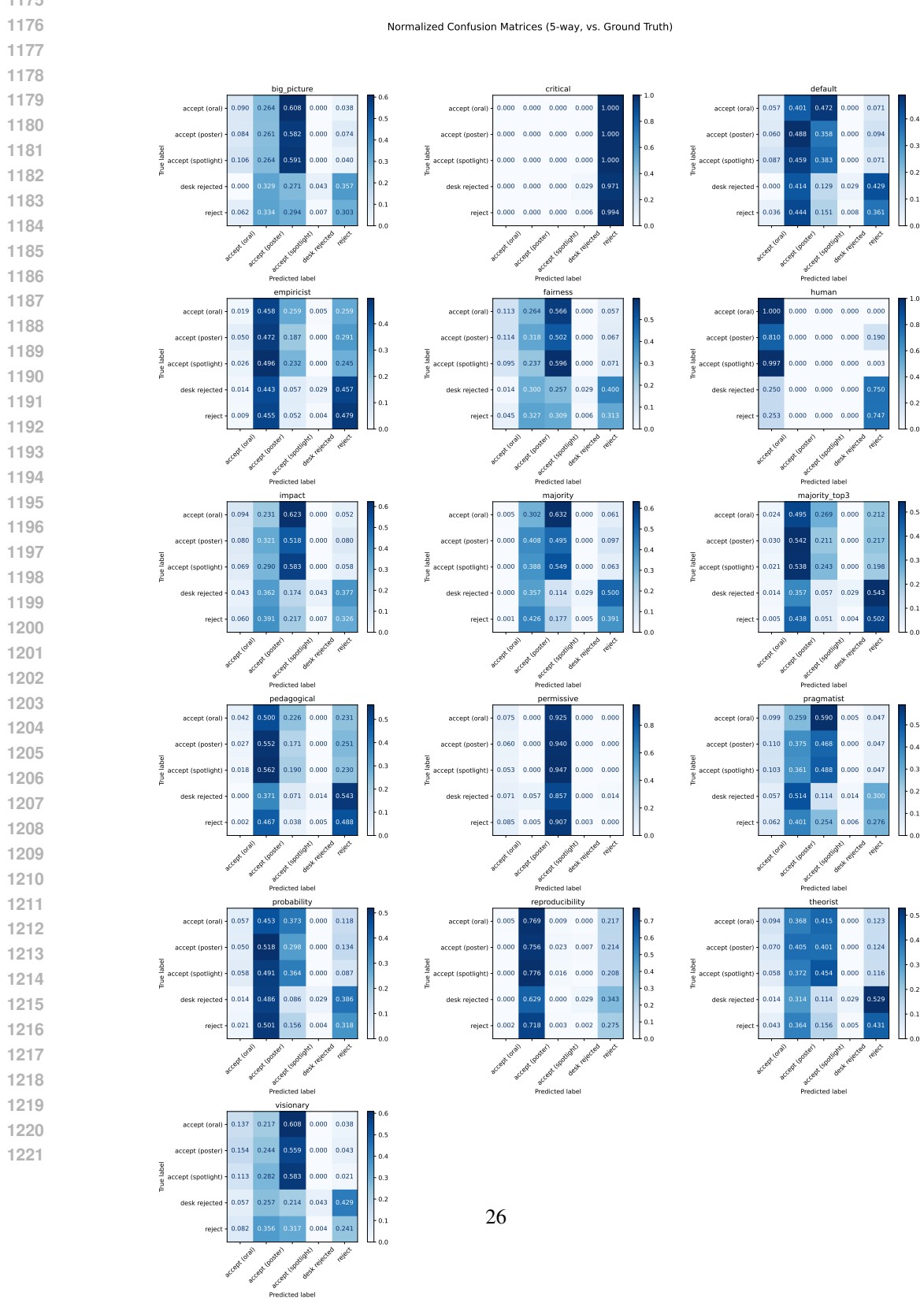

Figure 7: Confusion Matrices for 5-way Classification Task Post-Discussion

**Efficacy of AI Assistance**    Pilot studies have tested GPT-4 and other models in generating reviews, often finding them "slightly helpful" or capable of mimicking surface-level feedback, though lacking in deep critique (Robertson, 2023; Wu et al., 2023; Kadi & Ali Aslaner, 2024; Lo Vecchio, 2025). Specific case studies in medical and chemical domains suggest AI can serve as a "copilot" but not a replacement (Fuente-Ballesteros et al., 2025; Allibhai et al., 2025; Alshami et al., 2023). Meta-analyses of AI integration indicate a rising adoption despite these limitations (Carobene et al., 2024; Salman et al., 2025; Alchokr et al., 2024; Eger et al., 2025; Sun et al., 2024). Additionally, tools for positive reframing and annotation have shown promise in supporting the human workflow (Yang et al., 2025; D'iaz et al., 2024).

**Bias, Fairness, and Ethical Risks in Peer Review**    A critical motivation for *ReviewerToo* is addressing the inherent biases in human review while ensuring AI does not introduce new ones. The literature extensively documents prestige and affiliation bias in human review (Tomkins et al., 2017; Frachtenberg & McConville, 2022; Kulal et al., 2025; Ajani et al., 2025). Blind review processes are often compromised, and disparities persist (O'Connor et al., 2017; Conklin & Singh, 2022). Deploying AI judges introduces new fairness concerns, including position bias and verbosity bias (Shi et al., 2024b; Zhou et al., 2024a; Tripathi et al., 2025). Studies have shown that LLMs can identify gender disparities (Verharen, 2023) but may also hallucinate or act as silent judges with unacknowledged biases (Oriyad et al., 2025; Vasu et al., 2025; Zhang et al., 2022). Work on fairness extends to group fairness and preventing manipulation (Aziz et al., 2024; Jecmen et al., 2020; Huang et al., 2024).

**Ethical Challenges and Adversarial Attacks**    The integration of AI raises severe ethical questions regarding authorship and accountability (Solomon et al., 2023; Osmanovic-Thunström & Steingrimsson, 2023; Hiep, 2025; Kim, 2024; Alnaimat et al., 2025; Russo, 2021). Specific threats include hidden prompts injected into manuscripts to manipulate AI reviewers (Lin, 2025; Collu et al., 2025; Shi et al., 2024a; Keuper, 2025) and jailbreaking safety alignment (Mustafa et al., 2025; Andriushchenko et al., 2024; Kim et al., 2025a). Researchers have also highlighted the risks of sycophancy, where models agree with the user or author regardless of quality (Kim & Khashabi, 2025; Malmqvist, 2024; Atwell et al., 2025; Cheng et al., 2025; Arvin, 2025). Detection of AI-generated text and reviews remains a cat and mouse game, with watermarking and content-based detection proposed as mitigations (Rao et al., 2025; Yu et al., 2024; Chen et al., 2025b; Nemecek et al., 2025; Fraser et al., 2024; Guo et al., 2024).

## B.2    AUTOMATING THE PEER REVIEW WORKFLOW

A significant body of work focuses on the operational aspects of peer review, from reviewer assignment to decision support.

**Reviewer Matching and Editorial Support**    Finding qualified reviewers is a major bottleneck. AI systems for reviewer assignment and desk rejection prediction are well-studied (Alhoori et al., 2023; Farber, 2024; Lim et al., 2025; Anjum et al., 2019; Zhang et al., 2023b; Jakobsen & Rogers, 2022). Some propose market-based designs (Fernandes et al., 2025) or randomized assignments to mitigate manipulation (Jecmen et al., 2020). Prediction models for acceptance have utilized machine learning on text and metadata (Ghosal et al., 2018; Bao et al., 2021; Pandey et al., 2025; Hasan et al., 2024).

**Automated Review Generation**    Moving beyond prediction, recent work attempts to generate actual review content. Techniques range from retrieval-augmented generation (RAG) (Chitale et al., 2025; Taechoyotin & Acuna, 2025; Xu et al., 2024) to systems that simulate the entire review process (Cuaya-Simbro & Ruiz, 2025; Mahmoud et al., 2024; Zhu et al., 2025b; Lin et al., 2023; Zhuang et al., 2025; Zhou et al., 2024b). Tools like *ReviewAgents* and *ReviewWriter* aim to structure this generation (Gao et al., 2025; Su et al., 2025). While some models show promise in identifying critical problems (Zhang & Abernethy, 2025;

Li et al., 2025a), others highlight the risk of generic or hallucinated feedback (Zhu et al., 2025a; Ye et al., 2024).

**LLM Evaluation and the Judge Paradigm**   The credibility of *ReviewerToo* relies on the LLM-as-a-Judge paradigm, which has exploded in popularity. Researchers have established benchmarks to test LLMs as evaluators (Zheng et al., 2023; Hao et al., 2024; Chang et al., 2023; Yu et al., 2025). However, reliability varies, and design choices in prompts and references are critical (Yamauchi et al., 2025; Lee et al., 2024; Zhang et al., 2024; Li et al., 2024). Multi-agent debate and jury systems have been proposed to improve judgment quality (Bandi & Harrasse, 2024; Li et al., 2025c; Chern et al., 2024). Despite progress, issues with self-preference bias (Wataoka et al., 2024), style-over-substance (Feuer et al., 2024), and lack of robustness (Beyer et al., 2025; Guerdan et al., 2025) persist.

**Multi-Agent Simulation and Personas**   To address the monolithic nature of single-model reviews, multi-agent systems (MAS) and persona modeling are gaining traction. Frameworks like *TinyTroupe* and *AgentReview* simulate complex social dynamics and role-based interactions (Salem et al., 2025; Jin et al., 2024; Wang et al., 2025). These simulations allow for given-circumstance acting and debate (Xu et al., 2023; Li et al., 2025b). Research indicates that assigning specific personas (e.g., strict reviewer) can influence outcomes, though the persona effect requires rigorous quantification (Hu & Collier, 2024; Samuel et al., 2024; Kim et al., 2025c; Truong et al., 2025).

**The Future of Peer Review**   The field is moving toward hybrid, modernized workflows. Proposals include AI-enhanced scholarly communication platforms (Salih et al., 2025; Chen et al., 2025a; Nixon, 2024), decentralized blockchain-based systems (Bărbuţă & Alexandrescu, 2023; Gruendler et al., 2024), and open peer review models (Ross-Hellauer & Horbach, 2024; Tran et al., 2020). Theoretical work on social choice and mechanism design supports these innovations, aiming to align incentives and improve aggregation (Barcel'o et al., 2022; Kim et al., 2025b; Lee, 2023; Cohen-Addad et al., 2018; Tamblyn et al., 2023; Zhang et al., 2023a).

The literature reveals a critical juncture: while AI tools are proliferating, they are often deployed in fragmented or ad-hoc ways. *ReviewerToo* addresses the need for a modular, persona-aware framework that bridges the gap between automated efficiency and the nuanced, trusted judgment required for high-stakes peer review.

## C   LLM-AS-A-JUDGE PROTOCOL FOR ELO

We use the following update formula for ELO:

$$R'_A = R_A + K \cdot (S_A - E_A), \qquad E_A = \frac{1}{1 + 10^{(R_B - R_A)/400}},$$

where $R_A$ is the rating of system $A$, $S_A \in \{0, 0.5, 1\}$ is the observed score, and $K$ is the update constant. This produces a comparative ranking of review-writing quality across human and AI reviewers that integrates all five evaluation dimensions.

To ensure reliability and fairness in our LLM-based ELO evaluations, we use:

**Blinding.**   All reviews are anonymized prior to evaluation. System identities (e.g., "human," "persona X," "metareviewer") are removed, and formatting is standardized so that the judge cannot infer the source from stylistic cues.

**Randomization.**   For each pairwise comparison, the left/right order of reviews is randomized. The prompt to the judge LLM explicitly instructs it not to infer authorship based on order or style.

Table 3: Reviewer Persona ELO.

| Reviewer Persona | ELO$^{\uparrow}$ |
|---|---|
| big_picture | 364 |
| critical | 423 |
| permissive | 880 |
| reproducibility | 989 |
| default | 1136 |
| pedagogical | **1345** |
| pragmatist | 1182 |
| empiricist | **1558** |
| theorist | **1463** |
| visionary | 1097 |
| impact | 1121 |
| probabilistic | 1189 |
| fairness | 1154 |

**Outcome aggregation.**   The raw win/loss/draw outcomes are aggregated into ELO ratings using the logistic update formula described earlier in this section. For stability, we initialize all systems with identical ratings of 1,000 and use a moderate update constant ($K = 32$) for the first 30 matches of an agent, then reduced to $K = 16$ until the agent has played 500 matches, after which, it is fixed to $K = 10$. Final ratings are reported after convergence over the full set of pairwise matches.

**Match stratification.**   In large-scale settings, the number of possible review pairs can approach one million, which is computationally prohibitive. When fewer comparisons are run than the full set of possible matches, we employ a **stratified sampling strategy**: matches are distributed proportionally across (i) distinct query papers, and (ii) distinct parent review sources (e.g., human, persona, metareviewer). This ensures balanced coverage of both paper-level diversity and system-level diversity, while keeping the number of matches tractable.

**Quality control.**   A random subset of judgments (5%) is manually inspected by the authors to verify adherence to the evaluation rubric. Discrepancies between human inspection and the LLM judge are rare ($< 3\%$) and do not materially affect rankings.

**ELO Scores**

## D    QUALITATIVE EXAMPLES

---

**Paper 6Mxhg9PtDE (Real Decision: Oral) (Real Decision: Oral): Human Review**

**Summary:**
This paper demonstrates the shallow safety alignment issue through a variety of case studies. Essentially, the authors show that a variety of alignment attacks are successful because of a common issue within safety-aligned LLMS: only the first few output tokens are adapted during the model alignment process. Then the paper offers ways to mitigate this problem, which includes a data augmentation approach and a constrained optimization loss function.

**Strengths:**

- The paper is addressing an important problem, the vulnerability of safety alignment for LLMs, that can be very useful to real world problems.
- The paper ties together prior works in a way that makes it easier to learn from them (i.e. highlighting the common thread amongst successful alignment attacks: their exploitation of shallow safety alignment).

- The contributions of this paper lay the groundwork for future safety alignment solutions. They do offer a couple mitigation strategies, but exposing the shallow alignment issue could inspire many more mitigation approaches. It could also help us understand the success of other attacks and the success/failure of existing attack mitigation strategies.
- The paper includes a good variety of experiments (models, datasets, attacks types) and includes both empirical and theoretical support for their claims.
- The paper flows nicely. It is nicely organized. This makes the paper easy to follow and it makes the main point/contribution of the paper very clear.

## Weaknesses:

The explanation of related work is lacking. The related works are listed, but there is not much information that actually explains how your work differs from related work. For instance, you say "some works have also noted asymmetries..." But it would be nice to know how this differs from what you've observed. A lot of the statements you make about related work are very broad and could benefit from more detail. "Our work ties these potential failure modes... to potential shortcuts" - does your work do this for all pre-existing methods for improving alignment? Are there some failures that your work does not encapsulate? Also, you never seem to mention any solutions to these alignment failures. Are your methods (e.g. the data augmentation and constrained optimization) the only known mitigation strategies? If so, you should state this. If not, other mitigation strategies should be mentioned.

After applying your mitigation strategies, the ASR is still not zero and often isn't even that close to zero. This isn't ever really explained in the paper. You at one point say "the augmented model is still vulnerable...", but the paper would be stronger if you give more explanation. For instance, does the non-zero ASR mean that there is some other vulnerability apart from the shallow alignment issue? Or are your strategies just not fully fixing the shallow alignment problem?

Your contribution would be stronger if it were explained more clearly. When you say things like "this work is the first work attempting to consolidate the unified notion behind these attacks..." I don't quite understand what you mean. If other works have identified the shallow safety alignment effect, then what does it mean for you to "consolidate the unified notion"? Is shallow safety alignment a new term that you are introducing, because if so, I think you should make it more clear that you are introducing this new concept?

It is also hard to imagine this problem in a real-world setting/application. The paper would be stronger if, for example in the introduction, we were given an example of the effect that jailbreaks can have (e.g. him what scenario would some attacker be able to provide a deployed model with the start to a response)

## Questions:

- Why do you think the ASR still isn't 0 (and in many cases is not close to 0) after using your mitigation strategies?
- It seems like there could potentially be problems with the data augmentation approach since you are providing the model with these strange texts (e.g., you mention that the new texts are not coherent). Do you think that this matters? Is the model's learning going to be compromised when it is learning with these incoherent texts?

**Rating: 10: strong accept, should be highlighted at the conference**

---

## Paper 6Mxhg9PtDE (Real Decision: Oral): Empiricist

### Summary of Contributions

The paper introduces the notion of **shallow safety alignment**, i.e., that current alignment procedures for large language models (LLMs) mainly modify the probability distribution of the first few output tokens to induce a refusal prefix. The authors provide:

1. Empirical evidence (per-token KL divergence, token-wise loss and gradient analyses) that alignment effort is concentrated on early tokens.
2. A unified explanation of several known jailbreak and fine-tuning attacks as consequences of this shallow alignment.
3. Two mitigation strategies:
   - **Data-augmentation "deep safety alignment"** that mixes safety-recovery examples (harmful prefix → safe continuation) into continued fine-tuning.
   - **A token-wise constrained fine-tuning objective** that penalises deviation from the original aligned model more strongly on the first few tokens.
4. Extensive experiments on Llama-2-7B-Chat and Gemma-1.1-7B-IT showing reduced attack success rates (ASR) on prefilling, GCG, decoding-parameter, and fine-tuning attacks, while preserving utility on standard benchmarks (AlpacaEval, MMLU, BBH, MATH, GSM8K).

### Strengths

| Aspect | Comments |
|---|---|
| **Empirical rigor** | The paper systematically quantifies token-wise KL divergence and gradient norms, and links these to observed vulnerabilities. |
| **Unified perspective** | Provides a clear, parsimonious explanation that ties together a variety of jailbreak and fine-tuning attacks under a single mechanism. |
| **Mitigation proposals** | Both the data-augmentation and constrained-fine-tuning approaches are simple to implement and require only modest additional compute. |
| **Experimental breadth** | Evaluates on multiple models, several attack families, and a suite of utility benchmarks. |
| **Reproducibility details** | Appendices contain compute resources, optimizer settings, hyper-parameter values, and ablation studies. |
| **Ethical awareness** | Includes an ethics statement acknowledging the dual-use nature of the work. |

### Weaknesses

| Issue | Impact |
|---|---|
| Reliance on GPT-4 automatic judging | Safety evaluation depends on a single black-box judge; no human verification or inter-annotator agreement is reported, raising concerns about label noise. |
| Limited statistical analysis | Results are reported as mean±std over 3 runs (or 10 for GCG). No significance testing or confidence intervals; the small number of seeds may mask variability. |
| Scope of models and data | Experiments are limited to ~7B-parameter models and a modest augmentation set (256 triplets). Generalisation to larger models or different alignment pipelines (e.g., RLHF vs DPO) is not demonstrated. |
| Definition of "deep" alignment | The term is operationalised only through the proposed augmentation; alternative ways of deepening alignment (e.g., longer refusal prefixes, hierarchical prompts) are not explored. |
| Utility trade-off | While utility scores drop only slightly, the paper does not analyse qualitative impact on instruction-following; some degradation is visible (e.g., AlpacaEval loss from 51.8% to 49.5%). |
| Over-claiming | The conclusion that "future safety alignment should be made more than just a few tokens deep" is plausible but not fully substantiated beyond the attacks studied. |
| Ablation depth | The ablation on the number of "deep" tokens (parameter $C$) shows diminishing returns beyond ~100 tokens, but the paper does not discuss practical limits of how deep alignment can realistically be made. |
| Comparison to strongest baselines | The constrained fine-tuning is compared to "Standard SFT" and "Vaccine," but not to more recent jailbreak-defense techniques (e.g., self-refine, circuit-breakers). |

## Questions for the Authors

1. **Human verification** – Did you manually inspect a subset of GPT-4 judgements to estimate label accuracy? If not, could you provide a small human-annotated validation set?

2. **Seed sensitivity** – How much do the ASR numbers vary across random seeds (especially for the constrained fine-tuning)? Have you tried more than three repeats?

3. **Generalisation to larger models** – Do you have preliminary results on 13B or 70B-scale models, or on models aligned with pure RLHF rather than the proprietary pipeline used for Llama-2-Chat?

4. **Effect on non-harmful instruction following** – Beyond aggregated utility metrics, does the constrained objective affect the style or length of benign responses?

5. **Computational overhead in practice** – The constrained loss requires storing per-token probabilities of the original aligned model. How does this scale to models with >10B parameters and longer context windows?

### Suggestions for Improvement

1. Include a small human-rated safety benchmark to validate GPT-4 judgments and report inter-annotator agreement.

2. Increase the number of random seeds ($\geq 5$) and report confidence intervals or perform hypothesis tests to substantiate the observed reductions in ASR.

3. Test the methods on at least one larger model (e.g., Llama-2-13B-Chat) and on a model aligned via pure RLHF to assess generality.

4. Compare shallow vs. deep alignment under different training regimes (SFT only, RLHF, DPO) to clarify whether the phenomenon is specific to the proprietary pipeline.

5. Provide examples of benign responses before/after mitigation to illustrate subtle shifts in style or completeness.

6. Expand discussion on scalability of deep alignment (diminishing returns, token budget, impact on generation latency).

7. Release augmentation scripts and constrained-loss implementation to facilitate reproducibility.

### Overall Verdict

The paper presents a compelling empirical diagnosis of a concrete failure mode in current LLM safety alignment and offers two practical mitigation strategies that show measurable improvements across a range of attacks. While the experimental methodology could be strengthened (human safety validation, more extensive statistical analysis, broader model coverage), the contribution is novel, analyses thorough, and mitigation ideas of immediate practical interest to the community.

**Final Recommendation: Accept (Poster)**

## Paper 6Mxhg9PtDE (Real Decision: Oral): Theoretical

### Summary of Contributions

The paper introduces the notion of **shallow safety alignment** – the observation that current safety-aligned large language models (LLMs) primarily modify the generative distribution of only the first few output tokens to achieve refusal behavior. The authors:

1. Empirically characterize this phenomenon across Llama-2 and Gemma models, showing that KL-divergence between aligned and base models is concentrated in the early token positions and that prefixed refusal tokens dramatically reduce harmfulness.

2. Demonstrate that a variety of known jailbreak and fine-tuning attacks (adversarial suffix, prefilling, decoding-parameter exploits, and few-step harmful fine-tuning) can be explained as exploiting this shallow alignment.

3. Propose two mitigation directions:

(a) **Data-augmentation "deep safety alignment"** that augments training data with safety-recovery examples where the model must return to a refusal after a few harmful tokens.

(b) **Token-wise constrained fine-tuning** that penalises deviation from the original aligned distribution on early tokens via a novel regularised loss (Eq. 3).

4. Provide extensive experiments showing that both approaches improve robustness to the attacks above while preserving utility on standard benchmarks.

## Strengths

| Aspect | Comments |
|---|---|
| Novel conceptual framing | The paper formalises an intuitive observation (the "refusal-prefix shortcut") as *shallow safety alignment* and connects it systematically to a broad set of jailbreak and fine-tuning attacks. This unifying view is valuable for the safety community. |
| Comprehensive empirical analysis | Per-token KL, gradient-norm, and loss-norm studies across multiple models and datasets convincingly demonstrate the early-token bias. The prefixed-prefix experiments (Table 1) are simple yet powerful. |
| Concrete mitigation proposals | Both the data-augmentation and constrained-fine-tuning methods are easy to implement on top of existing pipelines, requiring only modest compute overhead (Table 12). |
| Thorough evaluation | The authors evaluate against a wide suite of attacks (prefilling, GCG, decoding-parameter, OOD code attacks, fine-tuning attacks) and report utility on AlpacaEval, MMLU, BBH, MATH, GSM8K. Ablation studies on hyper-parameters, $\beta$-schedules, and warm-up are included. |
| Theoretical grounding | Section F provides a clear derivation of the constrained loss, linking it to token-wise KL-regularised RL and to a soft-plus surrogate of a max-margin objective. Theorems 1–3 give a solid interpretation of the role of $\beta$. |
| Ethics discussion | The paper acknowledges that exposing failure modes can aid adversaries but argues the net benefit for safety research, satisfying the ICLR Code of Ethics. |
| Reproducibility | Detailed appendix sections (B–G) list compute resources, data construction, and hyper-parameters. Code-wise the methods are straightforward. |

## Weaknesses

| Issue | Impact |
|---|---|
| Limited scope of "deep" alignment | The data-augmentation approach only trains on a tiny synthetic safety-recovery set (256 examples). While it demonstrates feasibility, it is unclear how this scales to larger vocabularies, more diverse harmful content, or multi-turn dialogues. |
| Evaluation on a narrow model family | Experiments focus on Llama-2-7B-Chat and Gemma-1.1-7B-IT. It remains an open question whether the findings transfer to larger models (e.g., 70B) or to models with different fine-tuning pipelines (e.g., RLHF-only). |
| Reliance on GPT-4 as a safety judge | All safety metrics (harmfulness rate, ASR) are obtained via a GPT-4 classifier. Potential bias or miscalibration of this judge could affect conclusions; a human-in-the-loop validation would strengthen the claims. |
| Constrained loss hyper-parameter sensitivity | While ablations on uniform vs. biased $\beta$ are provided, selecting $\beta$ values for each token position may be non-trivial in practice. The paper does not propose an automated way to set $\beta$ (e.g., based on per-token KL statistics). |
| Utility degradation | Although utility drops are modest ($\approx$2–4% on AlpacaEval), the constrained loss sometimes harms downstream performance more severely (e.g., Table 4 shows utility loss on GSM8K). The trade-off analysis could be deeper. |
| Missing comparison to recent "circuit-breaker" defenses | The related work mentions short-circuiting (Zou et al., 2024) but does not empirically compare against it. A head-to-head would clarify the relative merits of the proposed methods. |

## Questions for the Authors

1. **Scaling of safety-recovery data** – How does performance vary when the safety-recovery set is enlarged (e.g., 1 k, 10 k examples) or when the harmful prefixes are sampled from real user queries rather than synthetic?

2. **Token-wise $\beta$ selection** – Did you try learning $\beta_t$ (or a schedule) from data (e.g., based on per-token KL divergence) instead of hand-crafting a step function?

3. **Multi-turn dialogues** – Does shallow alignment manifest similarly in multi-turn chat settings where the model can "recover" after a refusal in a later turn?

4. **Effect on non-refusal safe behaviors** – Some safety signals (e.g., content filtering via internal classifiers) are not captured by refusal prefixes. Does deepening alignment improve robustness to attacks that target those signals?

5. **Human evaluation** – Have you performed any manual verification of the GPT-4 safety judgments, especially for borderline cases (e.g., low-severity harmful content)?

6. **Compatibility with RLHF** – Can the constrained fine-tuning objective be combined with RLHF (e.g., as an additional KL term) without destabilising policy-gradient updates?

## Suggestions for Improvement

1. **Broaden the safety-recovery dataset** – Include diverse harmful continuations (different domains, code, multi-modal prompts) and evaluate the impact of dataset size on deep alignment.

2. **Automated $\beta$ scheduling** – Propose a simple heuristic (e.g., $\beta_t \propto \mathrm{KL}_t$) or a learnable gating network to set token-wise constraints, reducing manual tuning.

3. **Human-in-the-loop validation** – Sample a subset of outputs for each attack and have annotators verify the GPT-4 labels; report inter-annotator agreement.

4. **Compare with contemporaneous defenses** – Include baselines such as the "short-circuit" method (Zou et al., 2024) and the "Vaccine" approach (Huang et al., b) in the same experimental protocol.

5. **Evaluate on larger models** – Even a limited experiment on Llama-2-13B-Chat or a 70B model would strengthen the claim of generality.

6. **Ablation on multi-turn safety** – Test whether the deep alignment persists across conversational turns, perhaps using a dialogue benchmark (e.g., SafeChat).

7. **Improve manuscript readability** – Ensure all equations render correctly, add a schematic diagram of the shallow vs. deep alignment pipeline, and provide clearer captions for figures and tables.

**Overall Verdict**

The paper presents a **well-motivated, theoretically grounded, and empirically substantiated** analysis of an important safety failure mode in current LLM alignment pipelines. The notion of *shallow safety alignment* is novel and unifies a variety of jailbreak attacks, offering a fresh lens for the community. The mitigation strategies are simple, computationally inexpensive, and show promising improvements across a broad attack suite. While there are limitations concerning scalability, evaluation breadth, and presentation clarity, these do not undermine the core scientific contribution. **Final Recommendation: Accept (Spotlight)**

## Paper j7b4mm7Ec9 (Real Decision: Reject): Human Review

### Summary:
The paper provides a proper exploration of parameter lightening for watermarking models. The problem of mismatch between the actual decoding objective and the optimization objective of the commonly used decoding loss is solved. The solution to the above problem is attempted from the point of view of adding projection blocks and proxy losses. And the impact of each block on robustness at fine granularity is discussed after subdividing the watermarking framework.

### Strengths:
For the first time, we identify the mismatch between the optimization objectives of commonly used decoding losses (e.g., mean-square error and binary cross-entropy loss) and the actual decoding objectives, and confirm the existence of such a mismatch and its impact through ablation studies, which provides a new perspective for model optimization. The proposed separable projection head (PH) and decoding-oriented alternative loss (DO) effectively mitigate the negative impact of irrelevant optimization directions, allowing the lightweight model to achieve SOTA performance while maintaining high performance. The lightweight model outperforms existing models in terms of invisibility, robustness, and efficiency for other domains with limited computational resources, and minimizes performance loss by further compressing the model with a fine-grained deep watermarking framework.

### Weaknesses:
the authors don't seem to have considered the issue of capacity, and suggest discussing this part; there are some spelling problems, e.g. "robustness" in table9 at line 1005 we know that robustness depends mainly on the type and intensity of noise added during the training phase, i.e., the NWIP module, but in the The paper does not give detailed experimental parameters, but only describes the noise parameter settings in the testing phase.

### Questions:
In terms of comparison experiments, because the main concern in watermarking is capacity, imperceptibility, robustness and efficiency. In this paper, we mainly focus on the efficiency improvement, but we should ensure that the other parameters are the same for a fair comparison.HiDDen's algorithm can guarantee that the embedding in the grey scale image of size 1616 embedded in the length of 52 bits of information, the embedded information capacity can be up to 0.203 BPP. And the image used in this paper is a 3128*128 colour image, the embedded information length is 64 bits, the capacity is much lower than HiDDen's, so the result of such a comparison should be understood as unfair, please explain why. Besides, the embedding capacities of several other compared algorithms are not consistent, how are they compared?

Discriminative networks have been used in image watermarking frameworks since HiDDen and have shown advantages in enhancing the invisibility of watermarking frameworks. The authors of this paper did not consider this module in the newly proposed framework, please explain the reason, is it because high invisibility can be achieved without discriminative networks, to reduce the parameters so it is not used or there is another reason, please explain.

In response to the decomposition of the MSE loss function in the paper, we see the role of the different loss components, would like to ask if this decomposition is first proposed in this paper? Or is there already a readily available scheme in the field of knowledge distillation that performs a similar decomposition of the loss function, and this will affect the reassessment of the paper's innovativeness.

**Rating: 8: accept, good paper**

## Paper j7b4mm7Ec9 (Real Decision: Reject): Empiricist

### Summary of Contributions

The paper tackles the problem of building *lightweight* deep learning-based image watermarking systems that retain high invisibility and robustness. Its main contributions are:

1. **Analysis of the loss mismatch** – a decomposition of the commonly used MSE (and BCE) surrogate losses into *deflation*, *inflation* and *regularization* components, arguing that only the deflation term directly serves the decoding objective.

2. **Decoding-oriented surrogate loss (DO)** – a modified loss that retains the deflation term and restricts the inflation term to bits that lie within a "safe distance" $\varepsilon$ from the decision boundary.

3. **Detachable projection head (PH)** – an auxiliary projection module added only during training to absorb the inflation/regularization directions; it is removed at inference time.

4. **A five-module architectural framework** (image-preprocessing, message-preprocessing, feature-fusion, noised-image-preprocessing, message-extraction) that enables fine-grained ablations and module-wise pruning.

5. **A lightweight encoder/decoder** built solely from transposed-convolution and convolution layers ($\approx 16$ k parameters) that, when trained with DO or PH, achieves PSNR $\approx 41$–$42$ dB and average decoding accuracy $\approx 99.3$ % on a suite of combined distortions, outperforming several larger SOTA models (HiDDeN, MBRS, CIN, FIN) while using only $\sim 2$ % of their parameters.

### Strengths

| Aspect | Comments |
|---|---|
| Novelty of loss analysis | The explicit decomposition of MSE into deflation/inflation/regularization and the identification of "irrelevant" optimization directions is insightful and not previously articulated in watermarking literature. |
| Method simplicity | Both DO and PH are plug-and-play; they do not require architectural redesigns and can be applied to existing models. |
| Empirical performance | The lightweight model trained with DO/PH reaches higher PSNR than the strongest baselines (FIN) and matches or exceeds their robustness on the combined-noise benchmark. |
| Comprehensive ablations | The paper includes loss-component ablations, module-wise removals, PH block-number/channel-size studies, and hyper-parameter sweeps ($\varepsilon$, $\lambda$ weights). |
| Practical relevance | Reducing parameter count and FLOPs by $> 95\%$ is valuable for deployment on edge devices, a scenario explicitly motivated in the introduction. |
| Reproducibility cues | Public codebases for baselines are used; training details (optimizer, learning rate, hardware) are listed; all additional metrics (SSIM, LPIPS, $l_2$, $l_\infty$) are reported. |

## Weaknesses

1. **Statistical rigor** – All reported numbers are single-run averages; no confidence intervals, standard deviations, or significance tests are provided. Given the high accuracies ($\approx 99\%$), even small variances could alter the ranking against baselines.

2. **Limited attack spectrum** – Robustness is evaluated on six synthetic distortions (combined noise) and two diffusion-based attacks that are approximated by a Gaussian-noise + median-filter pipeline. Real-world attacks (cropping, rotation, scaling, aggressive JPEG with varying QF, format conversion) are missing, and the geometric-distortion results show the lightweight model lagging behind MBRS.

3. **Training overhead of PH** – While PH is removed at inference, its impact on training time, memory consumption, and energy is only qualitatively described ("inefficiency") without quantitative measurement. This makes it hard to assess the overall cost-benefit.

4. **Hyper-parameter sensitivity** – The safe distance $\varepsilon$ and the $\lambda$-weights strongly influence performance (Tables 18-19). The paper acknowledges manual tuning but does not provide a systematic tuning protocol or sensitivity analysis beyond a few discrete values.

5. **Theoretical justification** – The decomposition proof is presented, but there is no formal guarantee that minimizing DO leads to a tighter bound on decoding error than standard MSE/BCE, nor any convergence analysis for the PH-augmented training.

6. **Perceptual quality assessment** – PSNR, SSIM and LPIPS are reported, yet no user study or visual comparison beyond a single figure is provided; this limits confidence in the claim of "invisibility" for human observers.

7. **Clarity and presentation** – The manuscript contains many typographical artefacts (e.g., stray line numbers, inconsistent notation) that hinder readability and reproducibility.

## Questions for the Authors

1. **Variability** – How much does the decoding accuracy vary across random seeds and different train/validation splits? Could you provide standard deviations for the main tables?

2. **Training cost of PH** – What is the increase in GPU memory usage and wall-clock training time when PH is employed (e.g., for 4 projection blocks, 32 channels each)?

3. **Effect of $\varepsilon$ on different distortions** – Table 19 shows the safe distance tuned for combined noise; does the same $\varepsilon$ transfer well to single-distortion or geometric-distortion settings?

4. **Generalisation to other datasets** – Have you evaluated the model on datasets other than USC-SIPI (e.g., Kodak, DIV2K) to verify that the reported PSNR/accuracy are not dataset-specific?

5. **Ablation on loss weighting** – How sensitive are results to the choice of $\lambda_{DO_1}$, $\lambda_{DO_2}$ (or $\lambda_{PH_1}$, $\lambda_{PH_2}$)? Is there a principled way to set them?

6. **Comparison with recent lightweight baselines** – There are newer compact watermarking models (e.g., based on MobileNet or pruning techniques). How does your method compare to those?

## Suggestions for Improvement

1. **Statistical reporting** – Include at least three independent runs per configuration with mean ± std and perform paired statistical tests against baselines.

2. **Broader robustness evaluation** – Add experiments with geometric transformations (rotation, scaling, cropping) and with a range of JPEG qualities. If possible, test against state-of-the-art watermark removal attacks (e.g., GAN-based erasers).

3. **Training-efficiency analysis** – Quantify the additional GPU memory, FLOPs, and training time introduced by PH; possibly explore a lightweight PH (e.g., 1 block) to balance cost and gain.

4. **Hyper-parameter tuning protocol** – Provide a systematic search (grid or Bayesian) for $\varepsilon$ and $\lambda$ values, and report the resulting sensitivity curves.

5. **Theoretical bound** – Derive (or cite) a bound that relates the DO loss to the expected decoding error, clarifying why the reduction in inflation improves robustness.

6. **Human perceptual study** – Conduct a small user study (e.g., MOS) to corroborate PSNR/SSIM findings on invisibility.

7. **Manuscript polish** – Remove stray line numbers, ensure all equations are rendered, and unify notation (e.g., $L_{\text{deflation}}$ vs $L_{\text{deflation}}$).

**Overall Verdict**

The paper introduces a thoughtful analysis of loss mismatches in deep watermarking and proposes two practical techniques (DO and PH) that demonstrably improve a very compact model's performance. The empirical results are promising and the ablation studies are thorough, but the lack of statistical validation, limited attack diversity, and missing analysis of training overhead reduce the confidence in the claimed superiority. With the suggested revisions, the work would be a solid contribution to the ICLR community.

**Final Recommendation**

**Final Recommendation: Accept (Poster)** – the contribution is novel and empirically strong enough for inclusion, but further work is needed before it could be highlighted as an oral or spotlight paper.

---

## Paper j7b4mm7Ec9 (Real Decision: Reject): Theorist

### Summary of Contributions

The manuscript studies deep learning-based image watermarking with an emphasis on lightweight models. Its main contributions are:

1. **Identification of a "loss-mismatch"** – the authors argue that commonly used surrogate losses (MSE, BCE) contain components (inflation, regularization) that do not directly serve the decoding objective and thus waste capacity in small models.

2. **Two training-time remedies** –

   (a) **Detachable Projection Head (PH)**: an auxiliary head added only during training to absorb the "irrelevant" optimisation directions; it is removed at inference.

   (b) **Decoding-Oriented surrogate loss (DO)**: a reformulated loss that keeps only the "deflation" term and introduces a "safe-distance" hyper-parameter to limit the influence of inflation.

3. **A fine-grained five-module watermarking framework** (image-preprocess, message-preprocess, feature-fusion, noised-image-preprocess, message-extraction) that enables module-wise ablations and parameter reductions.

4. **A lightweight encoder-decoder architecture ($\approx$ 16 K parameters)** that, when trained with PH or DO, attains robustness and invisibility comparable to much larger SOTA models.

5. **Extensive empirical evaluation** on COCO/USC-SIPI, covering combined noise, diffusion-based attacks, geometric distortions, and knowledge-distillation baselines.

### Strengths

- **Practical relevance** – Reducing model size while preserving watermark robustness is an important engineering problem for deployment on edge devices, diffusion-based generative models, and neural-radiance-field pipelines.
- **Comprehensive experiments** – The authors evaluate many distortion types (six combined noises, diffusion attacks, geometric transforms) and report a wide range of metrics (PSNR, SSIM, LPIPS, accuracy). The ablation studies on individual modules and on the number of PH blocks/channels are thorough.
- **Plug-and-play nature** – Both PH and DO are described as modular additions that can be applied to existing lightweight watermarking pipelines without architectural changes.
- **Open-source spirit** – The paper mentions public code bases for baselines (HiDDeN, MBRS, CIN, FIN) and reports reproducibility details (datasets, optimizer, hardware).

### Weaknesses

1. **Theoretical novelty and rigor** – The decomposition of MSE/BCE into "deflation / inflation / regularisation" terms is essentially a re-parameterisation of the standard squared-error expansion; similar analyses exist in the learning-to-hash and metric-learning literature. The paper does not provide new theorems, nor does it prove that removing inflation/regularisation is *necessary* for lightweight models beyond empirical observation. The proposed DO loss is a heuristic truncation of the full surrogate loss plus a manually tuned safety margin $\epsilon$. No justification is given for the specific form of the safe-distance term, nor is there any analysis of its effect on the loss landscape (e.g., gradient bias, convergence guarantees). The PH module resembles an auxiliary classifier head often used for stabilising training (e.g., in deep metric learning). The manuscript does not situate this design within that broader context, making the claimed novelty ambiguous.

2. **Empirical methodology concerns** –
   - **Hyper-parameter sensitivity**: Both PH ($\lambda$ values, number of blocks, channel width) and DO ($\epsilon$, $\lambda$-weights) require extensive manual tuning. The reported gains are strongly dependent on these settings, raising concerns about reproducibility and fairness.
   - **Statistical significance**: Results are presented as single mean values; confidence intervals or multiple random seeds are absent. Given the modest absolute differences (often < 0.5 dB or < 1 % accuracy), it is unclear whether improvements are statistically robust.
   - **Baseline selection**: Comparisons exclude recent invertible or flow-based watermarking methods, which may impact the "state-of-the-art" claim.
   - **Training cost**: No quantitative measurement of training overhead is provided, though PH increases memory and computation.

3. **Presentation and clarity** – Numerous typographical errors, inconsistent notation, and low-quality figures hinder readability and verification.

4. **Ethical considerations** – While aligned with ethical standards, the lack of a dedicated "Broader Impact" section violates conference requirements.

### Questions for the Authors

1. Can you provide a formal analysis (e.g., a bound) showing that removing the inflation and regularisation terms reduces the required capacity for a given decoding error?
2. How sensitive is the final performance to the choice of $\epsilon$ across different noise levels? Is there an automated way to set $\epsilon$ without exhaustive search?
3. What is the additional GPU memory consumption and wall-clock time for training with PH compared to the baseline MSE training?
4. Have you evaluated the variance of PSNR/accuracy over multiple random seeds? Could you report 95 % confidence intervals for the key tables?
5. How do you envision the misuse of a highly robust lightweight watermark that could be embedded in malicious media (e.g., deepfakes)? Do you foresee any safeguards?

### Suggestions for Improvement

- Strengthen the theoretical contribution by linking the loss decomposition to calibration theory or re-framing it as an empirical design study.
- Automate hyper-parameter selection via validation-based scheduling for $\lambda$ and $\epsilon$.
- Report standard deviations or confidence intervals and conduct statistical tests.
- Include more recent baselines such as diffusion-based or invertible watermarking models.
- Improve manuscript clarity: fix typos, include key equations in main text, and enhance figure quality.
- Add a "Broader Impact" section discussing positive and negative societal implications.

### Overall Verdict

The paper addresses an important problem and provides extensive experiments, but lacks theoretical rigor, shows marginal gains, and suffers from clarity issues. Therefore, it does not meet ICLR 2025 acceptance standards. **Final Recommendation: Reject**

---

### Paper j7b4mm7Ec9 (Real Decision: Reject): Pedagogical

#### Summary of Contributions

The paper tackles the problem of building *lightweight* deep-learning based image watermarking systems that retain high invisibility and robustness. Its main ideas are:

1. **Identifying a loss-misalignment** – the commonly used MSE/BCE surrogate losses contain "inflation" and "regularisation" terms that do not directly contribute to decoding accuracy, leading to unnecessary parameter consumption in small models.

2. **Two training-time remedies** –

 (a) **Detachable Projection Head (PH)**: an auxiliary head is attached during training to absorb the irrelevant optimisation directions and is discarded at inference.

 (b) **Decoding-Oriented surrogate loss (DO)**: a reformulated loss that retains only the "deflation" term and limits the influence of inflation via a "safe distance" $\varepsilon$.

3. **A five-module decomposition of the encoder/decoder** (image-preprocess, message-preprocess, feature-fusion, noised-image-preprocess, message-extraction) that enables fine-grained ablations and parameter pruning.

4. **Extensive experiments** showing that a model with $\approx 0.02$ M parameters ($\approx 2$ % of SOTA) can achieve comparable or superior PSNR and decoding accuracy when trained with PH or DO, and that the methods are plug-and-play for other lightweight designs.

#### Strengths

| Aspect | Comments |
|---|---|
| Technical relevance | The problem of efficient watermarking is important for many downstream applications (e.g., diffusion models, NeRF). The identification of loss-misalignment is a useful observation that could inspire further work. |
| Empirical breadth | The authors evaluate on a wide range of distortions (combined noise, diffusion-based attacks, geometric transforms) and compare against several recent SOTA watermarking models. |
| Modular framework | The five-module decomposition is clearly motivated and the ablation tables (e.g., Table 7) illustrate its practical utility. |
| Plug-and-play claims | PH and DO are presented as methods that can be added to existing lightweight backbones without architectural changes – a potentially valuable engineering contribution. |

#### Weaknesses

| Category | Issues |
|---|---|
| **Clarity & Intuition** | • The core ideas (deflation vs. inflation) are explained only after a dense algebraic derivation that is largely hidden behind "Appendix A".
• The safe-distance $\varepsilon$ is introduced abruptly, without a visual illustration of its effect on the loss landscape. |
| **Narrative Flow** | • The paper jumps between high-level motivation, low-level algebra, and experimental details without smooth transitions.
• Section headings are all-caps and do not follow typical ICLR style.
• "Related Works" mixes historical and modern papers without a clear taxonomy. |
| **Presentation** | • Numerous line-numbers appear in text.
• Tables are inconsistently formatted.
• Figures are referenced but missing.
• Many typographical errors reduce professionalism. |
| **Accessibility** | • A newcomer would struggle to understand "deflation", "inflation", and "regularisation" without diagrams.
• Prior works are not clearly compared.
• Hyper-parameter tuning lacks guidance in main text. |
| **Reproducibility** | • No source-code link.
• Training details are scattered and incomplete. |
| **Overall readability** | The manuscript reads like a collection of logs; the abundance of numbers without interpretation hampers comprehension. |

## Questions for the Authors

1. Safe distance $\varepsilon$ – How is $\varepsilon$ chosen in practice? Would a validation curve help?
2. Projection head architecture – Please include a clear diagram of the projection block.
3. Training overhead – Quantify additional GPU memory and time for PH training.
4. Generalisation – Have PH/DO been tested on larger backbones (e.g., ResNet)?
5. Code release – Will the implementation be released for reproducibility?

### Suggestions for Improvement

1. Rewrite the core methodological section with diagrams and intuitive explanation.
2. Remove extraction artefacts and follow ICLR template.
3. Provide complete figures and well-formatted tables.
4. Add a concise "Practical Guide" with recommended values for $\varepsilon$, $\lambda_{PH}$, $\lambda_{DO}$.
5. Improve reproducibility by including implementation details and code link.
6. Strengthen related-work organisation into clear categories.
7. Thorough proofreading for grammar and typography.

### Overall Recommendation

The paper addresses an interesting niche (lightweight watermarking) and proposes novel loss-based ideas that yield strong results. However, communication quality is currently below ICLR standards—absent figures, fragmented narrative, and formatting artefacts make evaluation difficult. I recommend **rejection** and suggest substantial revision. **Final Recommendation: Reject**

# E  IMPLEMENTATION DETAILS

The system is implemented in Python with asynchronous orchestration and semaphores to control parallelism. All agents persist their outputs in a standardized directory layout, enabling caching, reproducibility, and downstream analysis. We use `gpt-oss-120b` as the primary large language model for all roles, served on 8xH100 GPUs with `vllm`. Reviewer prompts combine a base rubric with persona-specific instructions and the official reviewer guidelines from the ICLR website (including the code of ethics). [6] No fine-tuning is performed; all agents operate in instruction-following mode.

For training the XGBoost classifiers, we perform 5-fold cross validation and report the mean results across the 5 folds in Table 1. For XGBoost, we use 200 estimators, a max depth of 6, learning rate of 0.1, and for the Bert classifier, we finetune it for 20 epochs, use a learning rate of 2e-5, a batch size of 16. We use an 70-15-15 training-validation-test split and perform hyperparameter tuning on the validation set to test between learning rates {1e-5, 2e-5, 3e-5} and number of epochs in {3, 5, 10, 20, 30, 50}.

---

[6]https://iclr.cc/Conferences/2025/ReviewerGuide

| Agent | Responsibilities |
|---|---|
| **LitLLM** | • Retrieve and rank relevant papers
• Summarize top-$k$ works into concise review
• Provide grounding for reviewer, author, and metareviewer |
| **Reviewer** | • Paper summary
• Explicit strengths and weaknesses
• Checks: novelty, soundness, experiments, results/discussion
• Organization/presentation and impact
• Grounds judgments in manuscript or literature
• Categorical recommendation (Oral, Spotlight, Poster, Reject, Desk Reject) |
| **Author** | • Synthesizes rebuttal from reviews and literature
• Addresses criticisms, clarifies misunderstandings
• Proposes revisions (e.g., code release, ablations)
• Explicitly cites reviewer claims or literature |
| **Metareviewer** | • Summarizes reviewer stances and scores (pre-rebuttal)
• Identifies shared strengths and weaknesses
• Evaluates rebuttal effectiveness
• Tracks stance shifts post-rebuttal
• Highlights lingering concerns or disagreements
• Fact-checks reviewer claims; assigns significance scores
• Categorical recommendation |

Table 4: Overview of agents in the REVIEWERTOO pipeline and their responsibilities.

| Reviewer persona | Style and primary focus |
|---|---|
| **default** | Balanced, rubric-following reviewer aligned with ICLR 2025 guidance; covers soundness, novelty, impact, clarity without strong bias. |
| **critical** | Skeptical, flaw-finding stance; stress-tests novelty claims, methodology rigor, and baselines. |
| **permissive** | Supportive lens; highlights strengths and potential, assumes good faith, emphasizes positive interpretations of results. |
| **empiricist** | Evidence-first; scrutinizes datasets, baselines, metrics, statistical validity, and whether results support claims. |
| **pragmatist** | Real-world utility; feasibility, scalability, deployment cost, practitioner relevance, and adoption barriers. |
| **theorist** | Conceptual rigor; coherence and elegance of core ideas, logical soundness, evidence-theory alignment. |
| **pedagogical** | Communication quality; clarity, intuition, narrative flow, figure/table interpretability, accessibility to newcomers. |
| **big_picture** | Vision-first; long-term significance, paradigm-shift potential, conceptual promise over implementation details. |
| **reproducibility** | Replication rigor; missing hyperparameters, data splits, seeds, envs; checklist compliance and ambiguity removal. |
| **impact** | Foundations and representations; depth, interpretability, principles that advance long-term AI understanding. |
| **visionary** | Bold paradigm shifts and learning dynamics; speculative but plausible mechanisms and broader implications. |
| **fairness** | Practical elegance and scalability; efficient, implementable methods with robust large-scale validation. |
| **probabilistic** | Probabilistic rigor and generative modeling; uncertainty handling, principled inference, socially meaningful applications. |
| **metareviewer** | Synthesis and calibration; aggregates reviewers, evaluates rebuttal effectiveness, extracts/verifies facts, assigns significance, and produces AC-facing briefings and recommendations. |
| **majority** | A metareviewing baseline taking majority vote of all the *reviewer* agents. |

Table 5: Reviewer and metareviewer personas used in ReviewerToo and their primary emphases.

