# OpenReview forum: "ReviewerToo: Should AI Join The Program Committee? A Look At The Future of Peer Review"
_ICLR.cc/2026/Conference — Submitted to ICLR 2026_

### Official Review · Reviewer_h5Vu · 2025-10-15

**Soundness:** 3
**Presentation:** 3
**Contribution:** 1
**Rating:** 2
**Confidence:** 4

**Summary:**

This paper introduces a review system named ReviewerToo. The authors conducted detailed testing and analysis on a carefully curated dataset, ICLR-2k. After careful examination, I think that the paper lacks sufficient novelty.

**Strengths:**

The study is based on a carefully constructed, large-scale dataset, ICLR-2k, which contains 1,963 real papers stratigraphically sampled from the ICLR 2025 conference, ensuring the authenticity and representativeness of the experiments. The paper clearly validates the effectiveness of various components within the framework (e.g., conference guidelines, literature retrieval, author rebuttal phases), demonstrating that structured and contextualized information is crucial for improving the quality of AI reviews.

**Weaknesses:**

The paper overlooks a significant body of existing work on AI review. Many of its implementation details can be found in prior research, which the authors fail to discuss. This makes me question whether the paper's novelty and contributions are suitable for a venue like ICLR. For instance, ReviewerToo's external literature retrieval and novelty verification have already been implemented in DeepReview[1]. Another core contribution, the "diverse panel of personas," has been realized in AgentReview[2]. Furthermore, regarding the dataset, DeepReview[1] already provides complete review data for ICLR 2024 and ICLR 2025.

The paper's evidence does not adequately support the claim that ReviewerToo surpasses human review systems. For instance, the paper does not clarify how human scores were calculated. If the data is simply the original reviewer scores from ICLR and not from a new, controlled experiment, then the conclusion in line 694, "humans are highly effective at holistic judgments of paper quality," is unreliable. This is because the initial reviewer scores are causally linked to the final acceptance decision; for example, papers with all-positive scores are generally accepted.

Furthermore, I disagree with the claim in line 805 that the system "produced reviews often judged more constructive than the human average." The evaluation relies solely on an LLM-as-a-judge, which I believe is prone to significant bias. For example, it may favor longer or better-formatted (e.g., Markdown) reviews. These are known issues in LLM-as-a-judge evaluations, and a model like gpt-oss-120b, likely trained with extensive RLHF, would be predisposed to generating text that appeals to another LLM judge.


---

[1] DeepReview: Improving LLM-based Paper Review with Human-like Deep Thinking Process

[2] AgentReview: Exploring Peer Review Dynamics with LLM Agents

**Questions:**

How are the human scores calculated? Including F1, ELO, etc.

---

> ### Author Response · Authors · 2025-11-27
> **Author Response**
>
> Please find our response below:
>
> > The paper overlooks a significant body of existing work on AI review. Many of its implementation details can be found in prior research, which the authors fail to discuss...ReviewerToo's external literature retrieval...complete review data for ICLR 2024 and ICLR 2025.
>
> - Thanks for raising this issue. We have now added an extended literature survey in Appendix B. We have also run DeepReview-14B model and the prompt from Liang et al., (2023).
>
> | Reviewer Agent | Accuracy |
> | --- | --- |
> | Meta (all) (ours) | 81.8 |
> | DeepReview-14B | 62.5 |
> | Liang et al., (2023) | 71.0 |
>
> - **Differences from AgentReview**: While it utilizes multi-agent frameworks, they address fundamentally different research questions with distinct methodologies: AgentReview focusses on a sociological simulation designed to identify human-like biases (e.g., authority bias) and disentangle latent factors in the review process. In contrast, ReviewerToo is a technical utility study evaluating if AI can effectively perform the task of peer review with high accuracy (81.8%) against a human benchmark. Furthermore, the role of personas in both works are different, AgentReview uses personas to introduce variance and test system robustness against bias. ReviewerToo employs multi-agent debate not to simulate flaws, but as an optimization strategy to improve the consistency and quality of critiques on real-world tasks.
>
> **Differences with DeepReview**: The core component of the DeepReview dataset is its synthetic/simulated "structured intermediate review steps" and reviews, which were generated to *train* a model. We do not train a reviewer model, as we believe it is not the right approach to this problem -- we know human reviews are noisy. A modular framework that can be adapted or tweaked, makes it better experimental setup.
>
> > How are the human scores calculated? Including F1, ELO, etc.
>
> For the binary case, we treat ratings > 5 as accept and <=5 as reject. For the 5-way case, we use the following thresholds that were calculated based on the avg score of paper in each category in the dataset: {Accept (Oral): 7.8, Accept (Spotlight): 7.4, Accept (Poster): 6.05, Reject: 4.11, Desk Reject: 0.54}. Note that for 5-way classification, all reviewers for a given paper will be assigned the same label based on the avg human reviewer ratings. Our ELO computation is detailed in Appendix C and we follow the standard match simulation protocol to compute it -- we pit human reviews against llm reviews and update the ratings based on which review was better.
>
> > humans are highly effective at holistic judgments of paper quality," is unreliable.
>
> You are right. Human reviewers do have an unfair advantage in this case. We have removed that claim from the paper. Thanks for pointing this out.
>
> > I disagree with the claim in line 805 that the system "produced reviews often judged more constructive than the human average." The evaluation relies solely on an LLM-as-a-judge, which I believe is prone to significant bias. For example, it may favor longer or better-formatted (e.g., Markdown) reviews.
>
> This is a valid question. We tried "embellishing" human reviews with an LLM without changing the core content, and recomputed the ELO scores. The average human ELO in that case is still **938** compared and ELO of top-1% humans is 1474 compared to 1657 for the metareviewer agent.
>
> Liang et al., (2023). Can large language models provide useful feedback on research papers? A large-scale empirical analysis. https://arxiv.org/abs/2310.01783

---

### Official Review · Reviewer_7xzE · 2025-10-25

**Soundness:** 1
**Presentation:** 2
**Contribution:** 3
**Rating:** 4
**Confidence:** 3

**Summary:**

This paper proposes ReviewerToo, a modular framework for studying the use of AI-assisted peer reviews in AI conferences. To support this, the work introduces ICLR2K, a curated subset of the ICLR 2025 peer review dataset, where the paper primarily analyzes the predictive performance of acceptance decisions (2-way: accept/reject; 5-way: Oral, Spotlight, etc.). ReviewerToo consists of several agents (literature review agents, reviewer agents, meta-reviewer agents, and author agents) that coordinate to perform the review process and ultimately make decisions on papers. Notably, the reviewer agents adopt different personas (e.g., enthusiast, theorist) in an attempt to reflect the diverse perspectives of real-world reviewers. These reviewers utilize information from both the literature agent and the manuscript to make their decisions. Interestingly, aggregating the existing peer review protocols (e.g., including meta reviewer decisions) leads to improved performance.

**Strengths:**

- The paper proposes a modular framework that researchers working on AI-assisted peer review can experiment with, which I believe is a valuable contribution to the community.
- Many of the initial questions I had while reading the abstract were well addressed in the manuscript.
- The experiments with different personas and baselines were very interesting.
- Experiments and ablations were extensive.

**Weaknesses:**

- The proposed ICLR2K dataset consists only of ICLR 2025 review data. Why didn't the authors aggregate review data from different years or investigate whether the findings generalize across years?

- The main issue with this work is the lack of discussion and engagement with existing literature, as well as insufficient evidence to support some of the findings. For instance, the authors state that "Human reviewers exhibit low to moderate agreement with LLM reviewers (κ≈0.1–0.2), consistent with known levels of disagreement in real peer review" or claims like  "potentially sycophantic tendencies of LLMs." Where are the references? I understand that finding references for this type of work can be challenging, as much of the evidence is anecdotal, but the current paper includes only 8 references, which suggests that the authors have not invested sufficient effort in substantiating these claims with existing literature.

**Questions:**

**Suggestion**
- I believe the current topic of incorporating AI reviewers is controversial; as such, a more in-depth discussion of related works and supporting background regarding the use of such reviewers in AI conferences needs to be thoroughly addressed. The current work references only 8 papers, which is too few. There are numerous works that discuss such approaches, and this paper lacks sufficient engagement with the existing literature. I recommend the papers below and more extensive related works related to this manuscript.
-------
 - Is llm a reliable reviewer? a comprehensive evaluation of llm on automatic paper reviewing tasks
- Some ethical issues in the review process of machine learning conferences
- Position: The AI Conference Peer Review Crisis Demands Author Feedback and Reviewer Rewards
-  Position: The Artificial Intelligence and Machine Learning Community Should Adopt a More Transparent and Regulated Peer Review Process
 - Reviewergpt? an exploratory study on using large language models for paper reviewing
-----
- The visualization in Figure 1 can be improved. Please increase the font size of the numbers above the bars and provide an illustrative caption for the figure. Additionally, I believe Figure 1 does not need to be this tall—consider condensing its height.
- For Section 4.2, consider restructuring the content to directly indicate which part of each table corresponds to the explanation. For instance, Table 1 presents results in the order of ReviewerTooAgent followed by Supervised Baseline. However, Section 4.2 begins the explanation with Supervised Baseline, then abruptly transitions to Table 2 (which should also include a direct reference), before returning to Table 1. This organization makes the section difficult to follow.
- I am uncertain whether the confusion matrices in Figure 3 warrant a full page. There may be opportunities for more insightful analysis or discussion of related work.
- I recommend adding a one-sentence description of the ELO metric in the main manuscript, as this metric may be unfamiliar to most readers.
- Please remove the note in Table 4.
- When such persona is fixed, wouldn’t it be possible for some authors to game the reviewing system?, discussion regarding the weakness of the paper would be needed,

**Question**
- How did the authors select the top 1% of human reviewers in this paper's context?
- Wouldn't the ELO metric, which is based on LLM-as-a-judge, obviously favor reviews written by LLMs? LLMs naturally generate more detailed reviews than human reviewers, and as such, this does not seem to be an appropriate metric for understanding whether AI produces better reviews than humans.
- Is it possible to analyze the distribution of personas among human reviewers based on the personas set by the authors? Based on this distribution, the reviewer agents could adopt a similar distribution.

---

> ### Author Response · Authors · 2025-11-27
> **Author Response 1/2**
>
> Dear Reviewer 7xzE25
>
> Thank you for spending time to give feedback for our paper -- they were really helpful. Please find our response below:
>
> > Why didn't the authors aggregate review data from different years or investigate whether the findings generalize across years?
>
> The primary goal of this work is to systematically study the behavior of AI reviewers under controlled conditions; however, we are running parallel experiments with ICLR 2026 batch of submissions and are exploring other venues. Other reasons include availability of obtain a high‑quality, fully‑annotated set of human reviews (including meta‑reviews) that are publicly available on OpenReview – ICML and NeurIPS don’t have all the discussions public. Finally, we did not use past years data to avoid data contamination as gpt-oss might have been exposed to the submissions and their discussions from past years.
>
> > The main issue with this work is the lack of discussion and engagement with existing literature, as well as insufficient evidence to support some of the findings.
>
> Thank for pointing this out. We have now added an extended literature review in the Appendix B covering as many relevant papers as we could find. It contains references to works discussing the ethical issues in the review process of ML conferences and sycophantic behaviour of LLMs. Additionally, we now include some rebuttals written by the author agent showing hallucinations in those rebuttals, which also ultimately leads to higher false positive rates in reviewer agents (even in the presence of a final fact checking module).
>
> > For Section 4.2, consider restructuring the content to directly indicate which part of each table corresponds to the explanation. For instance, Table 1 presents results in the order of ReviewerTooAgent followed by Supervised Baseline. However, Section 4.2 begins the explanation with Supervised Baseline, then abruptly transitions to Table 2 (which should also include a direct reference), before returning to Table 1. This organization makes the section difficult to follow.
>
> You are correct, we have updated 4.2 and now mention all the ablation related things in a separate paragraph.
>
> > I am uncertain whether the confusion matrices in Figure 3 warrant a full page. There may be opportunities for more insightful analysis or discussion of related work.
>
> We have moved Figure 3 to the appendix.
>
> > I recommend adding a one-sentence description of the ELO metric in the main manuscript, as this metric may be unfamiliar to most readers.
> We have added a brief description of ELO now.
>
> > Please remove the note in Table 4.
> Thanks for catching this! We have removed it.
>
> > When such persona is fixed, wouldn’t it be possible for some authors to game the reviewing system?, discussion regarding the weakness of the paper would be needed,
>
> This is a good point. This is why we have multiple personas so that we can evaluate it from different perspectives.
>
> > How did the authors select the top 1% of human reviewers in this paper's context?
> The top-1% human reviewers are selected based on their ELO scores
>
> > Wouldn't the ELO metric, which is based on LLM-as-a-judge, obviously favor reviews written by LLMs? LLMs naturally generate more detailed reviews than human reviewers, and as such, this does not seem to be an appropriate metric for understanding whether AI produces better reviews than humans.
>
> This is a valid question. We tried "embellishing" human reviews with an LLM without changing the core content, and recomputed the ELO scores. The average human ELO in that case is still **938** compared and ELO of top-1% humans is 1474 compared to 1657 for the metareviewer agent. For our task, we believe we should be look at the ELO score ranges to assign a quality band to reviews instead of focussing on their absolute values to determine quality. For instance, the review quality of of the top-1% humans is the same as the metareviewer agent, even if there is ~200 ELO points gap; however, the gap between 900 to 1400 is too stark to make the same conclusion.

---

> ### Author Response · Authors · 2025-11-27
> **Author Response 2/2**
>
> > Is it possible to analyze the distribution of personas among human reviewers based on the personas set by the authors? Based on this distribution, the reviewer agents could adopt a similar distribution.
> This is a great point. We have been exploring prompt optimization techniques with DSpy and GEPA to figure out a prompt that achieves the human level performance, but so far, we have been unable to see any meaningful results. We have, however, performed llm-based clustering of different types of reviews (following the protocol from Liang et al., (2023)) and here are the results:
>
> | Label                               | Human n (%)        | ReviewerToo n (%)   |
> |-------------------------------------|---------------------|----------------------|
> | Implications of the Research        | 13,066 (14.64%)     | 41,976 (11.51%)      |
> | Add experiments on more datasets    | 6,369 (7.14%)       | 29,275 (8.03%)       |
> | Clarity and Presentation            | 15,330 (17.18%)     | 69,393 (19.02%)      |
> | Ethical Aspects                     | 1,134 (1.27%)       | 19,850 (5.44%)       |
> | Algorithm Efficiency                | 5,726 (6.42%)       | 22,878 (6.27%)       |
> | Reproducibility                     | 4,416 (4.95%)       | 59,265 (16.25%)      |
> | Comparison to Previous Studies      | 10,637 (11.92%)     | 25,547 (7.00%)       |
> | Theoretical Soundness               | 10,125 (11.34%)     | 31,259 (8.57%)       |
> | Missing Citations                   | 2,630 (2.95%)       | 942 (0.26%)          |
> | Add ablations experiments           | 2,635 (2.95%)       | 27,444 (7.52%)       |
> | Novelty                             | 17,181 (19.25%)     | 36,967 (10.13%)      |
>
> **Total (Human):** 89,249
> **Total (ReviewerToo):** 364,796
>
> Liang et al., (2023). Can large language models provide useful feedback on research papers? A large-scale empirical analysis. https://arxiv.org/abs/2310.01783
>
> We hope our response addresses your concerns and we are happy to discuss more if you have more questions.

---

> > ### Comment · Reviewer_7xzE · 2025-11-28
> > **From 8 references to 146 references?**
> >
> > I thank the authors for their reply. However, I am raising a serious concern about the expanded references.
> >
> > The original feedback was not about simply increasing citation count, but about insufficient engagement with relevant literature. The revision from **8 to 146 citations does not address this concern and in fact exacerbates it.**
> >
> > The new references in Appendix B is a bulk list without meaningful integration into the paper's narrative or argument. This approach:
> >
> > 1. Does not provide genuine engagement with prior work but simply list them out
> >
> > 2. Makes it difficult to understand which works are truly foundational or most relevant
> >
> > 3. Suggests the authors may not have thoroughly read or considered these sources
> >
> > 4. Seriously undermines scholarly rigor and reputation of ICLR conference.
> >
> > A strong related work section requires selective, thoughtful citation and engagement with relevant sections and discussion of prior work. I am really upset with what the authors have done. I am doing my best, spending my whole weekend to make a careful review, but the authors seem to be simply using LLMs to retrieve relevant works and list them as their related works. I hope the authors reflect on this.
> >
> > I am decreasing my score to zero.

---

> ### Author Response · Authors · 2025-11-28
> **On the increased #references**
>
> Dear Reviewer 7xzE27,
>
> We appreciate and truly value the time you are spending to give feedback for our paper -- it reflects in the rationality and quality of your comments.
>
> First, we want to address directly your impression that we "simply used LLMs" to generate a bulk list of citations. This is not what we did. We did not ask a general-purpose LLM to produce a literature review or fetch arbitrary references. We used a purpose-built, peer-reviewed LLM-based literature-search tool that has been developed over several years, benchmarked and designed to avoid hallucinations. The tool processed ~10,000 papers and identified 497 potentially relevant works. This took ~1 day. Notably, 201/497 of those papers were published in 2025 alone, 161 in 2024, and 53 in 2023 (with the rest spread between 2013 to 2022), which shows how the field of relevant work has exploded recently. We have spent more than a week after that just identifying which papers to include by compiling individual contributions from each paper and establishing how it relates to our work, to finally whittle it down to 150 (not 146) papers in total.
>
> After finalizing those papers, we have tried our best to convey a meaningful story to explain what is really happening in the field, starting with the community sentiments around the topic, then covering possible risks, biases, to finally discussing works on automating peer-review in specific. This whole process took around 8-9 days to complete and that extended lit review is not a result of some general purpose LLM outputting some text for our paper; we don't think LLMs are even capable of generating that many citations in the first-place.
>
> We want to also highlight that traditional citation practices can be biased by familiarity, venue visibility, or search-engine ranking. Tools powered by modern language models, when used responsibly, can help expand the search space, reduce these biases and select relevant papers more objectively. Sometimes we will get a lot of relevant results (like here), sometimes we will not (for instance, for a niche database paper, we only get about 50 relevant papers following the same process). All is to say that an LLM-powered tool does not diminish the researcher's role or responsibility for interpretation, curation, and synthesis, and evaluation should be based on whether those references are valid and correct, and not how many of them there are. We understand literature review is not simply a piece of text. And we have been very mindful about what to include in the paper.
>
> We are glad to receive a reviewer genuinely willing to spend time on our paper, so if you have any more questions or concerns, we are happy to address them.
>
> Thanks again for your continued engagement.

---

### Official Review · Reviewer_7g8L · 2025-10-31

**Soundness:** 2
**Presentation:** 2
**Contribution:** 1
**Rating:** 2
**Confidence:** 4

**Summary:**

Proposed a modular framework for deploying AI-assisted peer review.

**Strengths:**

Fine-grained analysis of model's prediction on different personas in prompt and methods.

**Weaknesses:**

[Goal of the work] I don't see any values of improving the acceptance predictions using AI, even though the AI system is more advanced with multi-agentic framework. Getting a higher prediction score means that current peer-review system should be replaced by your system or even better versions of it in the future?
[Technical merit] What is the core technical merit of this work? There are tons of multi-agent systems repliciating human scientists, reviewers, or any other stage of research. Is this work proposing yet another agentic pipeline? If so, the proposed framework seems too simple, linear-chained framework which was done by many prior work and other startups.

**Questions:**

See above

---

> ### Author Response · Authors · 2025-11-27
> **Author Response 1/2**
>
> Please find our responses below:
>
> > [Goal of the work] I don't see any values of improving the acceptance predictions using AI, even though the AI system is more advanced with multi-agentic framework. Getting a higher prediction score means that current peer-review system should be replaced by your system or even better versions of it in the future?
>
> We agree that we should not replace human decision‑making with an AI classifier. We also state this in the abstract (L013, L025), and other places in the manuscript like the discussion in Sec 6, that starts with exactly this point (L397).
>
> The prediction score is used only as an objective, reproducible proxy for measuring how well an AI reviewer aligns with the collective judgment of a real conference (i.e., the final meta‑review decision). This is a standard practice in the peer‑review literature (e.g.,  Hossain et al., 2024; Jin et al., 2025; Gao et al., 2025) where the agreement with ground‑truth decisions is reported to assess fidelity of an automated reviewer, not to propose a full substitution. In the paper, our stand is that there are some tasks that can be clearly off-loaded to an AI, while leaving the subjective parts to the human reviewer. It does not mean that the human reviewers don’t need to read a paper, they still do, but with better AI tools at their disposal, they can make better use of their time. Otherwise, reviewers are overstrained, and they either write low quality reviews that are super short, opinionated, and not helpful to the authors or they rely completely on AI to write long but superfluous reviews (which can be erroneous). This was also apparent with the recent macro-analysis around ICLR 2026 batch of submissions, highlighting more and more reviewers are relying on AI to write their reviews, but not necessarily in the right way. So, instead of stigmatizing the use of AI, we argue to build better AI tools to augment human reviewers, authors, and meta-reviewers, and improve the quality of science.
>
> Hossein et al., 2025. https://aclanthology.org/2025.naacl-long.395/
> Gao et al., 2025. https://arxiv.org/pdf/2503.08506
> Jin et al. 2025. https://aclanthology.org/2024.emnlp-main.70.pdf
> Zhu et al., 2025.
>
> > [Technical merit] What is the core technical merit of this work? There are tons of multi-agent systems repliciating human scientists, reviewers, or any other stage of research. Is this work proposing yet another agentic pipeline?
>
> We highlight our key contributions below compared to past works
>
> | **Novelty** | **Prior Work** | **Difference** |
> | --- | --- | --- |
> | **Flexible modular framework** that cleanly separates literature retrieval, persona‑conditioned reviewer generation, author rebuttal synthesis, and meta‑review aggregation. | Prior LLM‑review assistants (e.g., Idahl & Ahmadi 2024, Zhu et al., 2025) are monolithic agents that generate a single review without intermediate stages, and they only work with a single input. | Our design enables controlled experiments on each stage (e.g., ablations of literature grounding, persona conditioning, rebuttal handling, and conditioning on specific conference instructions). |
> | **Multiple explicit reviewer personas** (theorist, empiricist, pedagogical, critical, permissive, etc.) instantiated via prompt engineering and calibrated against conference guidelines. | Existing multi‑agent simulations (Anthis et al., 2025) model homogeneous agents; no work has systematically varied reviewing philosophy within a peer‑review workflow. | This allows us to quantify bias introduced by different reviewing styles and to study ensemble effects. We also systematically show the importance of generating diverse reviews. |
> | **Meta‑reviewer with fact‑checking module** that verifies reviewer claims against the manuscript and the retrieved literature before synthesizing a decision. | Most prior work treats the meta‑review as a simple majority vote. | Our fact‑checking reduces the impact of opinionated and less critical reviewer claims and yields a measurable improvement in decision fidelity. |
> | **Guidelines for responsible integration** derived from systematic bias analysis, not just anecdotal recommendations. | Many position papers (e.g.,  Latona et al., 2024) list high‑level recommendations. | Our guidelines are **data‑driven** (see Sec. 6.1) and include concrete protocol steps (e.g., persona‑mixing, rebuttal‑handling safeguards). |
> | **Reliable pipeline with open-weights models** | Prior works primarily study closed-source models like GPT, Claude, or Gemini family of models. | We use gpt-oss-120b for all our experiments and demonstrate that it is possible to get high-fidelity outputs by breaking down the complex task into simpler subtasks. |

---

> ### Author Response · Authors · 2025-11-27
> **Author Response 2/2**
>
> >  If so, the proposed framework seems too simple, linear-chained framework which was done by many prior work and other startups.
>
> While the high‑level flow (paper → literature → reviewer → rebuttal → meta‑review) may appear linear, the core of ReviewerToo lies in controlled, interchangeable modules that can be independently swapped, extended, or parallelized. Our framework runs parallel ensemble of 13 personas (theorist, empiricist, pedagogical, etc) that generate reviews simultaneously and feed them to the meta‑reviewer. This shows that the framework naturally supports non‑linear, multi‑agent configurations. Additionally, our work has several distinctions compared to prior work as noted previously. Finally, there is a need for the community to build benchmarks and robust/reproducible evaluations for this task and we hope our work pushes the needle in that direction.

---

### Meta-Review · Area_Chair_BsF6 · 2026-01-05

**Summary:**

This paper proposes ReviewerToo, a multi-agent framework for AI-assisted peer review. They did an empirical study on a curated ICLR 2025 subset (ICLR-2K). Reviewers generally acknowledge the effort to build a structured system and dataset, as well as the extensive experimental analyses, including persona-based reviewer agents and ablations of different review components. Reviewer 7xzE in particular finds the modular design potentially useful as an experimental platform for studying AI-assisted peer review, and both Reviewer 7g8L and 7xzE note that the persona-based analyses are interesting and thoughtfully explored. The dataset construction and validation of individual framework components are also seen as careful and systematic (Reviewer h5Vu).

**Reviewer Concerns:**

The primary weaknesses are limited novelty, unclear technical contribution, and insufficient engagement with prior work.
- Reviewer 7g8L questions the fundamental goal and value of improving acceptance prediction via AI, arguing that the framework appears to be yet another relatively simple, linear multi-agent pipeline without a clear technical advance.
- Reviewer h5Vu similarly emphasizes that many core components—literature retrieval, persona-based reviewers, and even comparable datasets which have already appeared in prior efforts (e.g., DeepReview, AgentReview), undermining claims of novelty.
- Reviewers 7xzE and h5Vu also highlight a serious lack of related-work discussion and citation support: key claims about reviewer agreement, sycophancy, and review quality are insufficiently substantiated, with only a small number of references despite a substantial existing literature.
- Methodological concerns further weaken the paper, including unclear definitions of “human scores,” potential circularity between reviewer scores and acceptance decisions (Reviewer h5Vu), reliance on LLM-as-a-judge metrics such as ELO that may systematically favor LLM-generated reviews (Reviewers 7xzE, h5Vu), and limited generalization due to focusing on a single conference year (Reviewer 7xzE). Additional concerns include presentation and organization issues, lack of discussion on system gaming and ethical implications, and overreliance on biased automatic evaluation.

**Reviewer Scores:**

- Reviewer 7g8L: Unchanged (Reject, score = 2). This reviewer’s concerns are fundamental and conceptual, focusing on the lack of clear motivation, unclear value of acceptance prediction, and absence of core technical novelty. These issues are unlikely to be resolved through discussion alone, and no other reviews substantially alleviate these concerns.

- Reviewer 7xzE: Unchanged (Weak Reject, score = 4). While this reviewer acknowledged strengths in modularity and experimentation, the discussion would likely reinforce concerns about insufficient engagement with prior literature, weak evidentiary support for claims, and methodological issues (e.g., dataset scope, LLM-as-a-judge bias). Given that multiple reviewers independently raised these points, the reviewer would likely converge toward a firmer reject.

- Reviewer h5Vu: Unchanged (Reject, score = 2). This reviewer strongly questions the novelty, correctness of claims comparing AI and human reviewers, and the validity of the evaluation methodology. Since these criticisms are substantive and echoed by others, discussion would likely confirm rather than soften their negative assessment.

---

### Decision · Program_Chairs · 2026-01-26

Reject